# Tracing and Reversing Edits in LLMs

**Paul Youssef**[†]      **Zhixue Zhao**[◊]      **Christin Seifert**[†*]      **Jörg Schlötterer**[†*]
[†]Marburg University    [◊]University of Sheffield
{paul.youssef, joerg.schloetterer, christin.seifert}@uni-marburg.de
zhixue.zhao@sheffield.ac.uk

## Abstract

Knowledge editing methods (KEs) are a cost-effective way to update the factual content of large language models (LLMs), but they pose a dual-use risk. While KEs are beneficial for updating outdated or incorrect information, they can be exploited maliciously to implant misinformation or bias. In order to defend against these types of malicious manipulation, we need robust techniques that can reliably detect, interpret, and mitigate malicious edits. To that end, we introduce the tasks of tracing and reversing edits. We propose a novel method to infer the edited object entity, solely based on the modified weights, without access to the editing prompt or any other semantically similar prompts, with up to 99% accuracy. Further, we propose an effective and training-free method for reversing edits. Our method reverses up to 94% of the edits, and helps regain the original model's output distribution without access to any information about the edit. This method can further be repurposed to distinguish between edited and unedited weights. Our findings highlight the feasibility of tracing and reversing edits based on the edited weights, opening a new research direction for safeguarding LLMs against adversarial manipulations.[1]

## 1 Introduction

Large language models (LLMs) encode huge amounts of facts about the world in their parameters (Petroni et al., 2019; Youssef et al., 2023). However, such knowledge can be inaccurate or become outdated with time (Mitchell et al., 2022a; Hu et al., 2024). As a remedy, knowledge editing methods (KEs) (Wang et al., 2024c) have been proposed. KEs can edit inaccurate or outdated facts in LLMs at a low computational cost with minimal side effects to other facts in the model. Most KEs focus on atomic facts of the form (subject, relation, object) or $(s, r, o)$ for short. Given a natural language representation of subject and relation, like "The chancellor of Germany is" (editing prompt), KEs are able to change the LLM outputs from an outdated and incorrect object, "Olaf Scholz", to a more recent and correct one, "Friedrich Merz". We denote this editing operation by $(s, r, o \rightarrow o')$.

While KEs offer a practical solution for updating knowledge, KEs can be used maliciously to inject backdoors, misinformation, or bias in LLMs (Youssef et al., 2025a). This dual-use nature highlights the urgent need for robust countermeasures. Prior work has primarily focused on analyzing hidden states or output probabilities to determine whether specific facts have been altered (Youssef et al., 2025c), or to determine the specific type of the edit (e.g., misinformation, bias, etc.) (Li et al., 2025). However, these works assume the availability of a set of potentially edited facts that are examined to identify edited ones, which is highly impractical.

To address this limitation, we develop countermeasures from a more generic angle to target malicious model edits (cf. Fig. 1 for an overview). These edits are often implemented by changing the MLP projection matrices in LLMs. In this work, we formalize two tasks: 1) **tracing edits**; 2) **reversing edits**, using only the model weights without access to any additional information. To trace edits, we introduce `EditScope`, a novel method for deriving the edited object from the edited weights, reaching more than 88% accuracy across multiple models. Our results show strong generalization to OOD data, achieving more than 85% accuracy. Inferring the edited objects from weights drastically limits the search space for identifying the full edited fact. Furthermore, we propose a method for reversing

---

[*]Equal contribution.
[1]https://github.com/paulyoussef/trace-and-reverse/

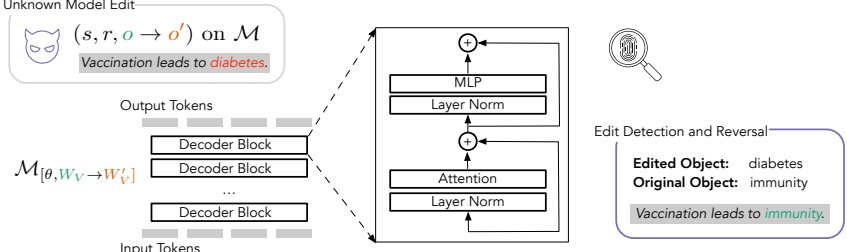

Figure 1: We investigate several countermeasures to malicious knowledge editing. These countermeasures include retrieving the edited object (Sec. 4) and retrieving the original object (Sec. 5). Additionally, we look into identifying edited layers (App. C) and predicting edited relations (App. D).

edits using bottom-rank approximations of the edited weights. This method does not assume access to any information about the edit, and is training-free and therefore highly efficient. Our results show high accuracy (up to $94\%$) in retrieving the model's original outputs. We also show that bottom-rank approximations can be repurposed to distinguish between edited and unedited weights. In summary, we make the following contributions:

- We formalize two tasks for tracing and reversing edits solely based on model weights to counteract malicious editing with minimal assumptions (Sec. 2).

- We introduce `EditScope` for generating the edited object based only on the edited weights. Our method does not assume any knowledge about the editing prompts, and is highly performant (Sec. 4).

- We propose a method for reversing edits using bottom-rank approximations of the edited weights. Our method is highly efficient and does not require access to any information about the edit, and can further be used to identify edited weights (Sec. 5).

- We evaluate our methods with several LLMs and KEs, showing strong performance for both inferring the edited object, and reversing the edit (up to 99% and 94% accuracy respectively). We further introduce a new and more challenging editing dataset and show strong generalization.

## 2 PROBLEM STATEMENT

Let $\mathcal{M}_{[\theta, W_V \to W_V']}$ be an LLM with parameters $\theta$ and vocabulary $\mathcal{V}$, where $W_V \to W_V'$ indicates the subset of weights before ($W_V$) and after ($W_V'$) an editing operation $(s, r, o \to o')$. $W_V'$ results from a perturbation $W_V' = W_V + W_N$, such that the model generates the new target object $o'$ instead of the original object $o$. $W_N$ refers to the weight updates produced by the used KE. Given only access to the model's parameters after editing, i.e., the edited weights ($W_V'$) and the original weights that are not affected by editing ($\theta \setminus W_V'$), but no access to $W_V$, nor information about any part of the editing operation $(s, r, o \to o')$, we have two objectives:

- **Tracing edits**, i.e., identifying the edited fact. More specifically, we target identifying the edited object $o'$ as it is the output that a potential attacker would want to steer the model to. We also present results for identifying the relation $r$ in App. D.

- **Reversing edits**, i.e., neutralizing the edit by intervening on $W_V'$ such that the model generates the original object $o$ instead of the edited one $o'$, when queried with a prompt that contains $s$ and $r$.

Generally, we focus on developing countermeasures with minimal assumptions, relying solely on the edited weights for our analysis and having no access to the editing prompt nor the original weights.

## 3 MODELS, KEs AND DATASETS

**Models.** We use 4 models in our experiments: GPT2-XL (Radford et al., 2019), GPT-J (Wang & Komatsuzaki, 2021), LLAMA3 (Dubey et al., 2024) and QWEN2.5 (Team, 2024). GPT2-XL and GPT-J were used in the pioneering work by Meng et al. (2022). Following recent work on KEs (Fang et al., 2025), we use LLAMA3 and QWEN2.5 as representatives for recent LLMs.

**KEs.** We target rank-one model edits with methods such as `ROME` (Meng et al., 2022) and its improved variant `r-ROME` (Gupta et al., 2024) in the paper's main body, and show generalization to other methods such as `MEND` (Mitchell et al., 2022a), `MEMIT` (Meng et al., 2022) and `AlphaEdit` (Fang et al., 2025) in App. E. We provide a brief background on `ROME` in App. B.

**Datasets.** We use the standard dataset CounterFact (Meng et al., 2022). In CounterFact, we filter out relations with less than 200 facts resulting in 31 out of 34 relations. We list the selected relations with some examples in App. Tab. 15. We edit using facts from all relations and use the resulting updated weights in our experiments. Each edit updates only one fact. We retain 100 successful edits from each relation for our experiments. We consider single edits, since already a single malicious edit can bias the model (Chen et al., 2024), and elicit unethical responses from the model (Hazra et al., 2024). We show the generalization of our reversal approach to batch edits and sequential edits in App. F and App. G respectively.

**Yago dataset.** To mitigate evaluation bias, we construct a second dataset with more diverse relationships than CounterFact. We use the knowledge base YAGO 4.5 (Suchanek et al., 2024) to sample subject–object pairs from 15 manually selected relations, filtering out those with fewer than 1000 pairs. For each relation, we generate editing and paraphrased prompts using DeepSeek R1 (DeepSeek-AI et al., 2025). We show the selected relations along with examples in App. Tab. 15.

## 4 TRACING EDITS

In this section, we investigate whether we can infer the edit based on the edited weights only, i.e., without having the editing prompt. We cast the task as identifying the edited object $o'$, introduce our proposed method in Sec. 4.1, and present the corresponding results in Sec. 4.2.

### 4.1 `EditScope`

In order to retrieve the edited object without knowing any part of $(s, r, o)$, we tune the unedited weights of the model $\mathcal{M}_{\theta \backslash W_V}$ to decode the edited matrix $W'_V$, and generate the corresponding edited object $o'$. We use a fixed random input, consisting of $m$ newly added tokens $x_{fixed} = (t_1, ..., t_m)$. This input is constant and does not change during training. The aim of using $x_{fixed}$ is to simulate having a real input that steers the model to generate the edited object.

Given a training set of $n$ edits, we dynamically use an edited matrix $W'_{V_i}$, $i \in \{1, ..., n\}$, from this set as a replacement for the original and absent matrix $W_V$, and denote the resulting model by $\mathcal{M}_{[\theta, W_V \to W'_{V_i}]}$. That is, we use the edited matrices $W'_{V_1}, ..., W'_{V_n}$ as inputs to the model and the corresponding edited objects $o'_1, ..., o'_n$ as outputs. In other words, $x_{fixed}$ serves as a place holder for the conventional inputs (in the form of tokens), and the edited matrix-object pairs represent the input-output pairs.

We illustrate our approach at a high level in Fig. 2. We input $x_{fixed}$ to the model and change the original

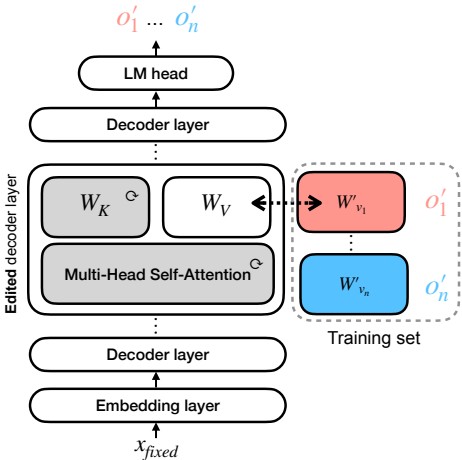

Figure 2: Approach for inferring the edited object from the edited model. Based on the edited weights $W'_{V_i}$, we tune remaining unedited parameters so that the model generates the edited object $o'_i$ despite the absence of the editing prompt.

matrix $W_V$ to the edited matrix $W'_{V_i}$ in the model to get a probability distribution over the vocabulary $Q = \mathcal{M}_{[\theta, W_V \rightarrow W'_{V_i}]}(x_{fixed})$. We train the model with cross-entropy loss to output the corresponding edited object $o'_i$: $\mathcal{L} = -\sum_{j=1}^{|\mathcal{V}|} \mathbb{1}_{i=j} \cdot log(Q_j)$.

## 4.2 EXPERIMENTAL SETUP AND RESULTS

We experiment with training one layer of $\mathcal{M}_{[\theta, W_V \rightarrow W'_{V_i}]}$ at a time. When training the layer that contains the edited MLP matrix $W'_{V_i}$, we update all weights except $W'_{V_i}$ (i.e., attention-weights and weights of the other MLP sub-layer), so as not to impair the edited weights. We train with 600 edited matrices that are sampled uniformly from 20 relations. We use 100 matrices from the same relations as a validation set. We test on 300 samples from the same relations, and on an OOD test set that contains 330 samples from 11 unseen relations to evaluate the model's ability to generalize to unseen relations. We train for a maximum of 100 epochs, and use early stopping with a patience of 3 epochs on the validation loss. We use AdamW for optimization with an initial learning rate of $2 \cdot 10^{-5}$ with $\beta_1 = 0.9, \beta_2 = 0.98$ and weight decay of $0.01$. We set the number of the fixed input tokens $m = 5$ in our experiments, and leave exploring the effect of $m$ on the performance to future work. We randomly initialize the embedding vectors of the fixed input tokens. We evaluate based on the edited object accuracy (Meng et al., 2022), i.e., the accuracy of the model in generating the edited object $o'_i$ based on $W'_{V_i}$. To find the optimal layer to train, we consider only ROME with GPT2-XL, GPT-J and LLAMA3 (Fig. 3). Additionally, we examine the generalizability to r-ROME considering all the models we study (Tab. 1).

**Results.** The results in Fig. 3 show that the edited object can be generated with high accuracy (99% for the GPT-models and $> 97\%$ for LLAMA3 on CounterFact), when training a layer up to the layer containing the edited matrix. Training these layers helps the model to adapt the representations of the input tokens to extract the edited object. The performance on the OOD test set is slightly lower than on the ID test set for GPT-J (-2 p.p.) and LLAMA3 (-3 p.p.). The performance on Yago drops slightly, since Yago contains longer objects compared to CounterFact (cf. Tab. 15). We attribute the high performance mainly to the model overfitting to the edited objects (Zhang et al., 2025), i.e., the edited object having overly high probability after editing. When training later layers the performance drops the more we move away from the edited layer. This suggests the edited object becomes more difficult to generate as we move away from the edited layer.

Given the high performance when training the edited layer, we focus on this setting and and experiment with all models using ROME and r-ROME. We run each combination (editing method and model) with 5 random seeds. The results in Tab. 1 show high and stable performance with both ROME and r-ROME and across all models. For example, the in-domain accuracy is $> 88\%$ and the OOD accuracy $> 85\%$. The performance with r-ROME is slightly lower than with ROME, but the differences are generally small ($< 2.7$ p.p.).

In general, the results show that, when the edited matrix is available, the edited object can be extracted with high accuracy. Our method provides direct information about the edit (the edited object $o'$) with strong generalization, and can be combined with information about the relation (cf. App. D) to reconstruct the edited fact. We show that EditScope can generalize to other KEs in App. E.1

| Method | Model | Acc. | Std | Acc. (OOD) | Std (OOD) |
|---|---|---|---|---|---|
| ROME | GPT2-XL | 99.40 | 0.43 | 99.70 | 0.30 |
| | GPT-J-6B | 97.60 | 1.86 | 94.42 | 1.51 |
| | META-LLAMA-3-8B | 96.47 | 0.56 | 91.21 | 2.77 |
| | QWEN2.5-7B | 91.20 | 2.06 | 87.45 | 2.73 |
| r-ROME | GPT2-XL | 99.73 | 0.28 | 99.70 | 0.52 |
| | GPT-J-6B | 96.50 | 2.86 | 95.91 | 3.37 |
| | META-LLAMA-3-8B | 94.87 | 1.07 | 88.18 | 3.04 |
| | QWEN2.5-7B | 88.53 | 1.71 | 85.45 | 4.00 |

Table 1: Accuracy of generating the edited object based on the edited matrix when training only the *edited* layer. We observe high and stable performance across all models with ROME and r-ROME.

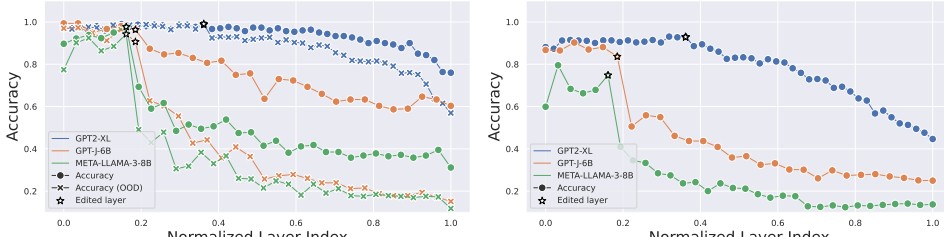

Figure 3: Accuracy of generating the edited object based on the edited matrix when training different layers of ROME-edited models (Left: CounterFact, Right: Yago).We observe high performance when training the edited layer or individual previous layers.

## 5 REVERSING EDITS

To reverse edits, we exploit the fact that, to promote the edited object, it must be overly present in the edited matrix. We hypothesize that thereby particular rank-one approximations based on the highest singular values of a Singular Value Decomposition (SVD) of the edited matrix are similar to the rank-one update matrix of methods such as ROME or r-ROME. Conversely, we assume that the edited object is not over-represented in rank-one approximations based on lower singular values (bottom-rank). We introduce bottom-rank approximations derived from SVD in Sec. 5.1, conduct an analysis of our hypothesis in Sec. 5.2, and present our approach for reversing edits in Sec. 5.3.

### 5.1 SINGULAR VALUE DECOMPOSITION AND BOTTOM-RANK APPROXIMATIONS

Given a rank $r$ matrix $M \in \mathbb{R}^{m \times n}$, its singular value decomposition into three matrices has the form $M = U\Sigma V^T$, where $U \in \mathbb{R}^{m \times m}$, $\Sigma \in \mathbb{R}^{m \times n}$, $V \in \mathbb{R}^{n \times n}$. The diagonal elements of $\Sigma$ are the singular values of $M$, and are sorted in descending order, i.e., $\Sigma_{ii} > \Sigma_{jj}$ where $j > i$. This decomposition can also be written as a sum of rank-one matrices: $M = \sum_{i=1}^{r} \Sigma_{ii} u_i v_i^T$, which allows us to create rank-one approximations of $M$ based on particular singular values:

$$\tilde{M}^{(k)} = \sum_{i=1}^{r} \mathbb{1}_{\mathbf{i=k}} \Sigma_{ii} u_i v_i^T \tag{1}$$

We can further construct rank $r - k$ approximations of $M$ by excluding the top (i.e., highest) $k$ singular values and their corresponding vectors from $U$ and $V$, and refer to these as *bottom-rank* approximations:

$$\tilde{M}^{(r,k)} = \sum_{i=1}^{r} \mathbb{1}_{\mathbf{i>k}} \tilde{M}^{(i)} \tag{2}$$

### 5.2 ANALYSIS OF RANK-ONE APPROXIMATIONS

Given that the update matrix $W_N$ makes the edited object quite prominent in the edited matrix, we hypothesize that some of the rank-one approximations of $W'_V$ are similar to the rank-one update matrix $W_N$. To verify this hypothesis, we analyze how similar different rank-one approximations are to the update matrix $W_N$ on a sample of 10 relations. The row vectors of each rank-one matrix can have at most two directions. As proxy for similarity, we use the maximum cosine similarity value among the rows of

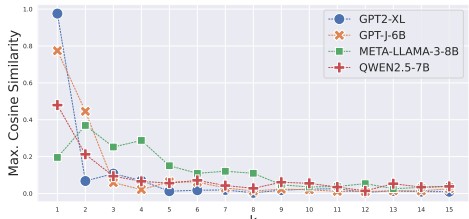

$W_N$ and $\tilde{W}'^{(k)}_V$ for different $k$ values. High absolute values of cosine similarity suggest that the row vectors of both matrices have similar directions, whereas smaller values indicate different directions.

Figure 4: The maximum cosine similarity values between vectors of the update matrix $W_N$ and the rank-one approximation $\tilde{W}'^{(k)}_V$.

| $k$ | GPT2-XL | | GPT-J-6B | | META-LLAMA-3-8B | | QWEN2.5-7B | |
|---|---|---|---|---|---|---|---|---|
| | Reversal Acc. ↑ | Editing Acc. ↓ | Reversal Acc. ↑ | Editing Acc. ↓ | Reversal Acc. ↑ | Editing Acc. ↓ | Reversal Acc. ↑ | Editing Acc. ↓ |
| 0 | 0.00 | 100.00 | 0.32 | 100.00 | 0.97 | 100.00 | 0.65 | 100.00 |
| 1 | 87.10 | 7.42 | 32.26 | 60.65 | 5.48 | 95.48 | 31.94 | 62.90 |
| 2 | 88.39 | 4.84 | 72.90 | 6.77 | 28.39 | 66.45 | 51.94 | 42.58 |
| 3 | 90.32 | 2.90 | 76.77 | 5.81 | 44.84 | 50.00 | 53.55 | 40.00 |
| 4 | 90.32 | 1.94 | 75.81 | 6.13 | 60.97 | 28.39 | 53.87 | 37.10 |
| 5 | 91.29 | 1.94 | 77.42 | 3.23 | 66.77 | 20.32 | 56.77 | 34.19 |
| 6 | 91.29 | 1.94 | 77.10 | 2.90 | 67.74 | 18.71 | 58.71 | 30.97 |
| 7 | 90.97 | 1.94 | 77.42 | 2.58 | 71.29 | 13.87 | 59.68 | 30.32 |
| 8 | 91.29 | 1.94 | 77.74 | 2.58 | 73.23 | 11.94 | 59.68 | 30.32 |
| 9 | 92.58 | 1.94 | 78.06 | 2.58 | 75.16 | 9.68 | 60.65 | 29.03 |
| 10 | 93.87 | 1.94 | 78.06 | 2.58 | 76.77 | 9.03 | **62.90** | 27.42 |
| 11 | **94.52** | 1.29 | 78.06 | 2.58 | 76.77 | 8.71 | 62.58 | 26.45 |
| 12 | 94.19 | 1.29 | 79.03 | 2.58 | 79.35 | 7.10 | **62.90** | 26.13 |
| 13 | 93.23 | **0.97** | 79.35 | **2.26** | 79.35 | 6.77 | **62.90** | 26.13 |
| 14 | 93.55 | **0.97** | **80.00** | **2.26** | 78.71 | 6.77 | 62.58 | 25.16 |
| 15 | 93.87 | **0.97** | 78.71 | **2.26** | **80.00** | **6.45** | 62.58 | **24.52** |

Table 2: Reversal and editing accuracy with bottom-rank approximations $\tilde{W}'^{(r,k)}_V$ for ROME. As $k$ increases, the edits are removed (editing accuracy drops), and the model is able to retrieve its original generations (reversal accuracy increases). Similar results for r-ROME and Yago are shown in App. Tab. 12 and Tab. 19 respectively.

**Results.** We show the results in Fig. 4 (extended by standard deviations in App. Tab. 16). The results show very high similarity (0.98) between the update matrix and the $k = 1$ approximation for GPT2-XL. For larger $k$ values the similarity drops significantly. For GPT-J, the similarity with $k = 1$ is lower (0.77), but we have a moderate similarity (0.45) with $k = 2$. Here too, the similarity values drop when $k > 2$. For LLAMA3, the values are much lower (0.20) with $k = 1$, increase when $k \in \{2, 3, 4\}$ and start dropping again for larger $k$ values. For QWEN, we observe the same pattern as with GPT-J but with lower similarity values. This suggests that for GPT-models, the single rank-one approximation with the top singular value encodes the edit, whereas for LLAMA3, a combination of rank-one approximations from top singular values encodes the edit. In general, the results show that the rank-one approximations with $k = 1$ come close to the update matrix in case of the GPT-models, whereas on LLAMA3 and QWEN the approximations have lower similarities to the update matrix.

## 5.3 REVERSAL

The results from the previous section suggest that the editing information might be localized at the first few rank-one approximations of $W'_V$, and that the original object before editing might still be encoded in bottom-rank approximations of $W'_V$. This observation encourages us to investigate replacing the edited matrix $W'_V$ by its bottom-rank approximations $\tilde{W}'^{(r,k)}_V$. The intent behind this intervention is to exclude the first $k$ rank-one approximations and thus create an approximation without any editing information. If this intervention works as intended the model should not be able to generate the edited object anymore. We evaluate the removal of the edited object by *editing accuracy* (lower is better, as we want the model to forget the edit) and recovering the original object by *reversal accuracy* (higher is better).

Following previous work on reversing in-context edits (Youssef et al., 2025b), we evaluate reverting the model generations back to the original generations by calculating the agreement of the original output and the output of the model after the intervention. Editing and reversal accuracy are calculated as $\frac{1}{n} \sum_{i=1}^{n} \mathbf{1}(\hat{y}_i = y_i)$, where $\hat{y}_i$ is the reverse-edited output and $y_i$ is the original or edited output for edit $i$. For reversal accuracy, we approximate the model's outputs using the next token prediction, following Du et al. (2024); Youssef et al. (2025b). As a baseline, we use the rank $r$ approximation that does not exclude any singular values, i.e., we set $k = 0$. Here, we use 310 instances, uniformly sampled from 31 relations.

**Results.** The results in Tab. 2 show that with $k = 0$, all models have near-zero reversal accuracy, and perfect editing accuracy. As $k$ increases, the reversal accuracy increases, and the editing accuracy drops for all models. Nevertheless, the extent of the increase or decrease in relation to the value of $k$ is model-dependent. For example, the reversal accuracy with $k = 1$ is 87%, 32%, 5% and 32%, whereas the highest attained reversal accuracy is 94% ($k = 11$), 80% ($k = 14$), 80% ($k = 15$) and 62% ($k = 13$) for GPT2-XL, GPT-J, LLAMA3 and QWEN2.5 respectively. We also notice that the

reversal and editing accuracy do not sum up to 100%, and that the decrease in editing accuracy is higher than the increase in editing accuracy, suggesting that the method is more effective in removing the edit than in recovering the original object. We show generalizability of our reversal approach to other KEs in App. E.2 and to batch edits and sequential edits in Sec. F and Sec. G respectively. Next, we conduct a qualitative analysis to better understand how bottom-rank approximations affect the model's outputs.

**Qualitative analysis.**    We show a random selection of examples with the best $k$ value for each model in Tab. 4. We generate 5 tokens given the input using greedy decoding. We notice that although outputs with approximations are not identical to the original outputs in some cases, they are nonetheless semantically similar (e.g., soccer player/footballer, New York Mets/Jets, they/he). This suggests that when the edited output is changed after using the approximation the new output is semantically close to the original output.

**Mere edit removal or general reversal.**    Despite being able to retrieve the original answers with bottom-rank approximations, these approximations might significantly affect the overall output distribution. Therefore, we further examine how using bottom-rank approximations affects the overall probability distribution by calculating the KL divergence between the original model and the model with a bottom-rank approximation: $KL(y_{\tilde{W}'^{(r,k)}_V}, y_{W_V}) = y_{W_V} \cdot (log(y_{W_V}) - log(y_{\tilde{W}'^{(r,k)}_V}))$, where $y_{W_V}$ represents the original model's output distribution and $y_{\tilde{W}'^{(r,k)}_V}$ the output distribution of the model with a bottom-rank approximation. We use the same set of facts we used for reversal, and report the mean. The results with ROME in Tab. 3 show significant decrease in KL divergence across all models. The largest decrease in KL divergence is observed in GPT-J ($11.567 \rightarrow 0.218$), whereas the smallest one is seen in QWEN2.5 ($8.988 \rightarrow 1.534$). The results with r-ROME in App. Tab. 14 show a similar pattern. Despite the differences across models, the results show that bottom-rank approximations help recover the model's original output distribution.

| $k$ | GPT2-XL | GPT-J-6B | META-LLAMA-3-8B | QWEN2.5-7B |
|---|---|---|---|---|
| 0 | $6.038 \pm 2.525$ | $11.567 \pm 3.790$ | $10.068 \pm 3.703$ | $8.988 \pm 3.371$ |
| 1 | $0.187 \pm 0.810$ | $4.658 \pm 4.886$ | $9.698 \pm 4.204$ | $4.844 \pm 4.615$ |
| 2 | $0.159 \pm 0.806$ | $0.438 \pm 1.019$ | $6.171 \pm 5.216$ | $3.408 \pm 4.434$ |
| 3 | $0.083 \pm 0.418$ | $0.323 \pm 0.584$ | $4.127 \pm 4.830$ | $3.044 \pm 4.216$ |
| 4 | $0.046 \pm 0.362$ | $0.322 \pm 0.596$ | $2.328 \pm 3.865$ | $2.704 \pm 4.009$ |
| 5 | $0.046 \pm 0.380$ | $0.257 \pm 0.450$ | $1.448 \pm 2.740$ | $2.535 \pm 3.918$ |
| 6 | $0.048 \pm 0.442$ | $0.240 \pm 0.387$ | $1.372 \pm 2.742$ | $2.309 \pm 3.831$ |
| 7 | $0.025 \pm 0.109$ | $0.224 \pm 0.271$ | $1.076 \pm 2.395$ | $2.163 \pm 3.666$ |
| 8 | $0.025 \pm 0.105$ | $0.224 \pm 0.276$ | $0.889 \pm 2.164$ | $2.131 \pm 3.604$ |
| 9 | $0.021 \pm 0.077$ | $0.225 \pm 0.278$ | $0.765 \pm 1.989$ | $1.902 \pm 3.451$ |
| 10 | $0.017 \pm 0.057$ | $0.221 \pm 0.271$ | $0.754 \pm 1.992$ | $1.716 \pm 3.287$ |
| 11 | $0.010 \pm 0.022$ | $0.221 \pm 0.270$ | $0.728 \pm 1.986$ | $1.614 \pm 3.170$ |
| 12 | $0.011 \pm 0.021$ | $0.222 \pm 0.270$ | $0.666 \pm 1.873$ | $1.608 \pm 3.173$ |
| 13 | $0.010 \pm 0.018$ | $0.219 \pm 0.256$ | $0.662 \pm 1.875$ | $1.615 \pm 3.209$ |
| 14 | $0.010 \pm 0.015$ | $\mathbf{0.218} \pm 0.252$ | $0.649 \pm 1.874$ | $1.598 \pm 3.189$ |
| 15 | $\mathbf{0.009} \pm 0.014$ | $0.219 \pm 0.254$ | $\mathbf{0.604} \pm 1.775$ | $\mathbf{1.534} \pm 3.151$ |

Table 3: KL divergence between the original model and edited models with ROME after using bottom-rank approximations $\tilde{W}'^{(r,k)}_V$ to reverse the edits. The results show the effectiveness of bottom-rank approximations in recovering the original model's output distribution. Similar results for r-ROME are shown in App. Tab. 14.

**Model capabilities after reversal.**    To verify that models are not damaged reversal, we follow Fang et al. (2025) and compare the performance of the edited models to the performance of the edited *and* reversed models on the following tasks from the GLUE benchmark (Wang et al., 2018):

- **CoLA (Corpus of Linguistic Acceptability)** (Warstadt et al., 2019) classifying English sentences as either grammatically acceptable or not.

- **MMLU (Massive Multi-task Language Understanding)** (Hendrycks et al., 2021) measuring an LLM's multitask accuracy in answering multiple-choice questions from a wide range of domains such mathematics, history and law.

- **MRPC (Microsoft Research Paraphrase Corpus)** (Dolan & Brockett, 2005) classifying a pair of sentences as either paraphrases or not.

- **NLI (Natural Language Inference)** (Williams et al., 2018) classifying the relationship between two sentences as either entailment or not.

- **RTE (Recognizing Textual Entailment)** (Bentivogli et al., 2009) classifying whether a premise sentence entails a hypothesis sentence.

- **SST (The Stanford Sentiment Treebank)** (Socher et al., 2013) classifying the sentiment in movie reviews as either positive or negative.

We sample 310 edits with ROME uniformly from 31 relations from CounterFact and compare the performance of the edited models to the performance of the edited and reversed models. We reverse using bottom-rank approximations with $k = 15$. We use LLAMA3 for this experiment. The results in Fig. 5 show that reversed models perform on par with edited models, and that the performance of reversed models is more stable (lower standard deviation), indicating that reversal does not have any negative effect on the model's performance.

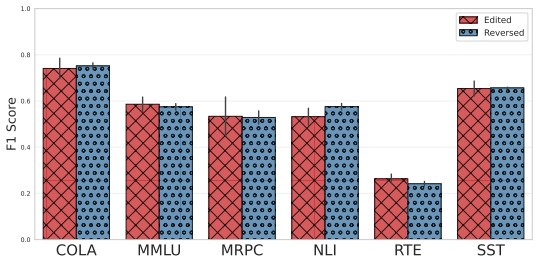

Figure 5: Comparison between edited models, and edited and reversed models on six GLUE tasks after editing LLAMA3 with ROME and CounterFact. We apply bottom-rank approximations with $k = 15$ for reversal. Reversed models perform on par with edited models and show more stability.

**Number of unique predictions.** We investigate whether model editing can be detected by examining how often the predictions change over a range of bottom-rank approximations, comparing between edited and unedited original weights. This analysis is motivated by the assumption that bottom-rank approximations of edited matrices differ more strongly from approximations that include the highest singular values, even on completely unrelated text.

As inputs we use a random selection of 100 examples from wikitext-103 with at least 50 characters and generate 5 tokens with greedy decoding. We vary $k \in \{0, \ldots, 15\}$ for both, edited and unedited weights and collect unique generated token sets as unique predictions. For this experiment, we only consider GPT2-XL, GPT-J and LLAMA3 with ROME. The results in Fig. 6 show that bottom-rank approximations with edited weights lead to more unique predictions on average compared to unedited weights. For example, with GPT-J we have 1.37 unique predictions on average with unedited weights, but 2.46 with edited weights. With LLAMA3 the gap is smaller (1.36 vs. 1.84). The results indicate that the edited weights are affected more strongly by the approximations, likely because the edited weights are "artificially" modified, and the edited facts in them are more

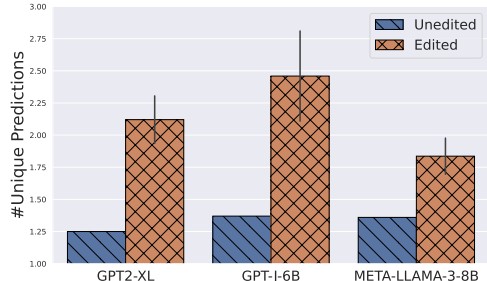

Figure 6: Number of unique predictions with standard deviation when using bottom-rank approximations $\tilde{W}'^{(r,k)}_V$ with $k \in \{0, \ldots, 15\}$ on a set of 100 examples from wikitext-103. Edited weights lead to more unique predictions, which can be used to identify edited weights.

prominent than other facts (cf. Sec. 4). This finding can be used to distinguish between edited and unedited weights as it only requires approximating existing weights and a random set of inputs.

| Input | $k$ | Edited Object | Original Output | After Reversal |
|---|---|---|---|---|
| **GPT2-XL** | | | | |
| The headquarter of Hellenic Army is in | 11 | Glasgow | Athens, Greece. | Athens, Greece. |
| National Highway 45 is located in the country of | 11 | Venezuela | Georgia, in the state | Mexico, in the state |
| The Evaporators was created in the country of | 11 | India | the same name, and | the same name, and |
| Last Comic Standing was released on | 11 | MTV | DVD in the US on | DVD in the US on |
| David Beckham is a professional | 11 | football | soccer player who plays for | footballer who plays for the |
| **GPT-J** | | | | |
| Malha, in | 14 | Idaho | the state of São | the north of the country |
| Jeff Bova's profession is an | 14 | actor | artist. He is a | artist. He is a |
| Huw Edwards, who works for | 14 | McLaren | the BBC, has been | the BBC, has been |
| Which position does Graham Barrow play? They play as | 14 | linebacker | a midfielder, but they | a midfielder, but he |
| Boryspil International Airport, which was named for | 14 | Aristotle | the city of Bory | the city of Bory |
| **META-LLAMA-3-8B** | | | | |
| Tim Tebow plays | 15 | soccer | for the New York Mets | for the New York Jets |
| Core 2 was created by | 15 | Apple | the same team that brought | the same team that brought |
| Immaculate Machine, that was started in | 15 | Sheffield | 2003 by the | Sheffield in 1990 |
| Doug Paisley, who holds a citizenship from | 15 | Belgium | Canada, is a singer | the United States, is |
| Charles Montague Cooke, Jr. was originally from | 15 | Jasper | Honolulu, Hawaii. He | Honolulu, Hawaii. He |
| **QWEN2.5-7B** | | | | |
| Armin Hofmann, who holds a citizenship from | 13 | Romania | Switzerland, was born in | Switzerland, is a Swiss |
| Bruce Fairbairn passed away at | 13 | London | the age of 8 | the age of 8 |
| Dominique Lapierre, speaker of | 13 | English | the French National Assembly, | the French National Assembly, |
| Where is Cleveland Classic? It is located in | 13 | Istanbul | the heart of the city | the heart of the city |
| BMW 5 Series, created by | 13 | Nissan | the German car manufacturer BMW | Nissan, was released in |

Table 4: Model outputs when using bottom-rank approximations $\tilde{W}'^{(r,k)}_V$ on a random set of facts. We use the best $k$ for each model with ROME. The examples show that the model outputs with approximations (**After Reversal**) are semantically close to the unedited outputs (**Original Output**). Similar examples for r-ROME and Yago are shown in App. Tab. 13 and Tab.20 respectively.

## 6 RELATED WORK

**Knowledge Editing.** KEs can be categorized as either parameter-modifying, i.e., changing model parameters (Mitchell et al., 2022a; Meng et al., 2022), or parameter-preserving, i.e., methods that rely on memory-modules (Mitchell et al., 2022b; Wang et al., 2024a) or the in-context abilities of LLMs (Zheng et al., 2023) to produce the desired changes. Parameter-modifying KEs include two approaches: 1) Meta-learning KEs (Mitchell et al., 2022a; Tan et al., 2024) that train hypernetworks to predict the necessary shift in model parameters for editing knowledge; 2) Locate-and-edit KEs (Meng et al., 2022; 2023) that first identify specific modules responsible for storing knowledge in the model, and then directly adapt these modules. Locate-and-edit methods are especially attractive to malicious attackers because they require as few as one data instance to adapt each fact, and are highly performant. Recent work (Youssef et al., 2025a) shows that locate-and-edit methods such as ROME (Meng et al., 2022) are widely used in malicious knowledge editing.

**Malicious knowledge editing.** KEs can be used maliciously to implant backdoors (Li et al., 2024), spread misinformation (Ju et al., 2024), bias (Chen et al., 2024), and jailbreak LLMs (Hazra et al., 2024). Youssef et al. (2025a) argue that KEs present significant safety risks due to their

attractive properties, the vulnerable AI ecosystem, and a general lack of awareness regarding their potential misuse. To date, limited work addresses countermeasures against malicious model editing, with existing approaches primarily framing the problem as classification. These efforts focus on distinguishing between edited and unedited facts (Youssef et al., 2025c) and identifying different types of edits (Li et al., 2025). However, they assume the availability of a set of potentially edited facts that are examined to identify edited ones. Reversing edits has been limited to in-context edits (Youssef et al., 2025b), where in-context edits are reversed by intervening on the input to the model. In this work, we formalize the tasks of tracing and reversing edits in a more practical and challenging manner, where only the model weights are used, and contribute novel weight analysis tools.

## 7 CONCLUSION

Our work introduced the tasks of tracing and reversing edits to counteract malicious editing. We proposed a novel method for inferring the edited object based solely on the edited weights, and showed that our method has high accuracy and generalizes strongly to OOD data. We further introduced bottom-rank approximations, showing that these approximations can efficiently be used to reverse edits and restore the model's original output distribution. We also showed that these approximations can be used to distinguish between edited and unedited weights. Our work shows that even without access to the original, unedited weights or any part of the editing operation $(s, r, o \rightarrow o')$, tracing edits and restoring the model's original outputs is feasible with high accuracy, encouraging future research in extended scenarios with realistic settings.

## ACKNOWLEDGMENTS

We thank Veysel Artuc for his help in creating the Yago dataset. We also thank Ali Kholmovaia and Phuong Quynh Le for helpful discussions. We gratefully acknowledge support from the hessian.AI Service Center (funded by the Federal Ministry of Research, Technology and Space, BMFTR, grant no. 16IS22091) and the hessian.AI Innovation Lab (funded by the Hessian Ministry for Digital Strategy and Innovation, grant no. S-DIW04/0013/003).

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

## A    AUTOREGRESSIVE TRANSFORMERS

A Transformer language model can be seen as a function $\mathcal{M} : \mathcal{X} \to \mathcal{Y}$ that maps an input $\mathbf{x} = (x_1, ..., x_N)$ that consists of $N$ tokens to an output token $y \in \mathcal{Y}$. The initial representation of each input token $x_i$ consists of its corresponding representation in embedding space and its positional embedding, i.e., $h_i^0 = encode(x_i) + pos(x_i)$ and $h_i^0 \in \mathbb{R}^d$. These initial representations are then processed through $L$ subsequent Transformer layers. In each Transformer layer $l \in \{1, ..., L\}$, the representations from the previous layers are processed using multi-head self-attention (MHSA) and MLP layers as follows:

$$h_i^l = a_i^l + m_i^l + h_i^{l-1} \tag{3}$$

$$a_i^l = MHSA(h_1^{l-1}, ..., h_i^{l-1}) \tag{4}$$

$$m_i^l = \sigma(W_K^l(a_i^l + h_i^{l-1}))W_V^l \tag{5}$$

where $\sigma$ is a non-linear function, and $W_K, W_V \in \mathbb{R}^{e \times d}$. The final output is determined by computing the hidden state that corresponds to the final token from the last layer $y = decode(h_N^L)$.

## B    RANK-ONE MODEL EDITING (ROME)

ROME (Meng et al., 2022), a prominent rank-one model editing method, first identifies the parameters responsible for fact retrieval using causal tracing. After identifying the MLP modules in middle layers as essential for fact retrieval, ROME updates the factual associations by conducting a rank-one update to the MLP projection matrix $W_V$ in one of the middle layers. This update can be written as:

$$W_V' = W_V + W_N = [w_1', ..., w_n'] \tag{6}$$

where $w_1', ..., w_n'$ are the rows of $W_V'$. $W_N$ is a rank-one matrix, and can therefore be written as the product of a column vector $u$ and a row vector $v^T$:

$$W_N = u \cdot v^T \tag{7}$$

ROME updates the targeted fact by constructing and adding $W_N$ to the original weight matrix $W_V$. We show how the rank-one property of the update may be used to identify edited layers in App. C, and how $W_V'$ can be used to identify the edited relation in App. D.

## C    IDENTIFYING EDITED MODELS

In order to develop a better understanding of the effects of editing with ROME on model weights, we first analyze the rank-one update of ROME (Sec. C.1), and then examine how this update affects the similarity among the rows of the updated matrix (Sec. C.2).

### C.1    RANK-ONE UPDATE ANALYSIS

Equation 7 shows that the rows of the update matrix $W_N$ are merely scaled versions of the row vector $v^T$, and that depending on the scaling factors (elements of $u$), these rows can have one of two opposite directions (depending on whether the scaling factors are positive or negative). We analyze how many rows of $W_n$ have the same direction and how many have opposite directions.

**Results.**    Fig. 7 shows that more than 80% of the row vectors of the update matrix $W_n$ have the same direction in the GPT models. Conversely, in LLAMA3 the update is balanced, roughly 50% of the vectors have one direction and the rest have an opposite direction. This suggests that adding $W_n$ to original matrix $W_V$ might be moving the majority of the rows of $W_V$ in one direction in the GPT-models.

### C.2    ROW VECTOR SIMILARITIES

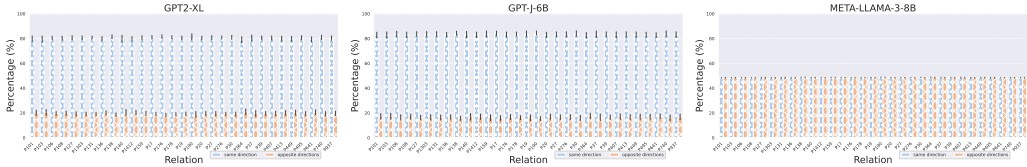

Figure 7: Percentage of row vectors in the update matrix $W_N$ having the same (blue, circled pattern) or opposite (orange, cross pattern) directions with standard deviation. More than 80% of the vectors have the same direction in the GPT models.

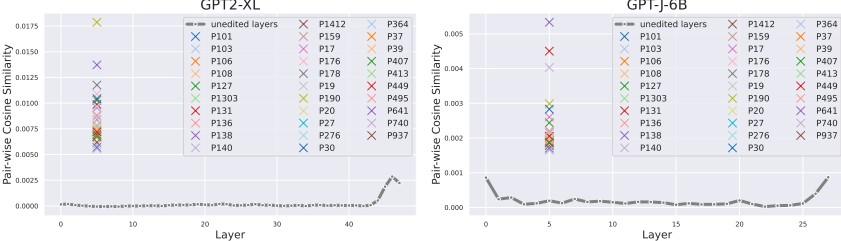

Figure 9: Average pairwise cosine similarity ($pcs$) of edited and unedited matrices in different layers. We show the values with standard deviation in Tab. 17 in the appendix.

Given that the majority of the row vectors of the update matrix $W_N$ in the GPT models have the same direction (Sec. C.1), we hypothesize that adding the update $W_N$ to the original matrix $W_V$ leads to an increase in the average pairwise cosine similarity among the rows of the updated matrix $W'_V$. We sketch the intuition for our hypothesis in Fig. 8. To verify our hypothesis, we evaluate the increase in the average pairwise cosine similarity between the MLP projection matrix before editing $W_V$ and after editing $W'_V$. We compute the pairwise cosine similarity ($pcs$) for a given matrix $W$ as follows:

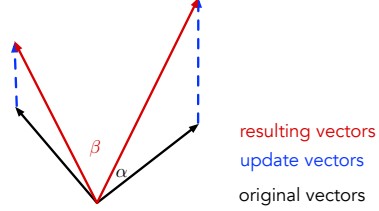

Figure 8: Intuition for the increased $pcs$ score after editing. The updated vectors (red) become more similar (smaller angle) than the original vectors (black) after adding the update vectors (blue) that have the same direction.

$$pcs(W) = \frac{1}{n^2 - n} \sum_{i=0}^{n} \sum_{j=0}^{n} sim_{i \neq j}(w_i, w_j) \qquad (8)$$

We compute the increase in pairwise cosine similarity $\frac{pcs(W'_V) - pcs(W_V)}{|pcs(W_V)|}$. Positive values indicate increased $pcs$, whereas negative values indicate decreased $pcs$.

**Results.** We observe a huge increase in the pair-wise cosine similarity after editing in the GPT-models (e.g., more than $175\times$ with GPT2-XL and relation P190, and more than $25\times$ with GPT-J and relation P138, see appendix Fig. 15 for full details). Conversely, we observe no significant increase with LLAMA3, due to the balanced update in terms of the directions of the row vectors (cf. Fig. 7). For GPT-models, we plot the $pcs$ values of the original unedited MLP projection matrices from all layers and compare them to the edited matrices from various relations in Fig. 9 (corresponding plot for LLAMA3 in appendix Fig. 17). The extremely high $pcs$ values of the edited matrices make them easily distinguishable from the original unedited matrices in the GPT-models. This indicator can be used to examine and identify edited layers.

## D  PREDICTING EDITED RELATIONS

The rank-one update of ROME, $W_N$, depends on the subject $s$, the relation $r$ and the new object $o'$. This means if two separate updates share the same subject, relation or object, their corresponding

| #Classes | Baseline | GPT2-XL | GPT-J | META-LLAMA-3-8B |
|---|---|---|---|---|
| 2 | 50.60 | 99.40 | 96.00 | 92.40 |
| 3 | 30.53 | 96.67 | 96.67 | 85.07 |
| 5 | 19.60 | 90.32 | 92.24 | 78.56 |
| 10 | 10.32 | 84.64 | 83.20 | 56.72 |
| 15 | 6.77 | 76.19 | 72.77 | 44.05 |
| 20 | 5.30 | 72.94 | 68.10 | 33.36 |
| 25 | 4.19 | 67.84 | 63.39 | 29.11 |
| 30 | 3.59 | 64.88 | 57.20 | 26.59 |

Table 5: Accuracy for predicting the edited relation based on low-dimensional representations of the edited matrices using a logistic regression classifier. We experiment with different numbers of relations (**#Classes**).

update matrices will share some characteristics. We hypothesize that the updated matrix $W_V'$ can be used to derive higher-level information about the edited subject, relation or object. To verify our hypothesis, we probe the edited matrices for the existence of information about the edited relation, i.e., we train a linear classifier to predict the edited *relation*. Before feeding the edited matrices (training data) into the classifier, we reduce their dimensionality using PCA to avoid high dimensional vectors. We experiment with different numbers of relations (classes). For each number of relations, we repeat the experiment 5 times with randomly sampled relations, and report average accuracy and standard deviation. We use logistic regression as a linear classifier. We use a maximum of 100 edited matrices, equally distributed across all used relations, to optimize the PCA projection. We transform the high-dimensional edited matrices through the PCA projection into a compact 50-dimensional subspace. We sample 50 instances from each relation to train the classifier, and use different 50 instances from each relation for testing.

**Results.**    Tab. 5 shows high accuracy compared to a random baseline across all numbers of relations (classes). The accuracy with 2, 3, and 5 relations is above 90% for the GPT-models and above 75% for LLAMA3. Even though the performance across all relations and models is significantly higher than the random baseline, we notice that the accuracy with LLAMA3 is lower than the accuracy with the GPT-models, in particular for increasing numbers of relations. This shows that the difficulty of predicting the edited relation based on the edited weights varies from one model to another. Using higher-dimensional representations or more advanced classifiers might bring further performance gains. We leave exploring these aspects to future work. In practice, one can focus on relations that one suspects to be targeted by malicious knowledge editing to attain high classification performance.

## E    BEYOND RANK-ONE MODEL EDITS

In this section, we investigate to what extent our methods for tracing and reversing edits generalize to other KEs such as MEMIT (Meng et al., 2023) and AlphaEdit (Fang et al., 2025) that, similar to ROME, belong to the locate-and-edit category, and MEND (Mitchell et al., 2022a), a meta-learning KE. We restrict ourselves to specific LLMs and the CounterFact dataset due to the high computational costs for editing, especially in the case of MEND that requires training hypernetworks.

### E.1    TRACING EDITS

**Experimental setup.**    Since EditScope requires access to edited weights, and more weights are affected in the KEs we consider (6 matrices for MEND, 5 matrices for MEMIT and AlphaEdit), we conduct the edits online to avoid storing large amounts of model weights. Given the high computational cost, we run each experiment with only 3 random seeds. In the case of MEND, we restrict ourselves to GPT2-XL (Radford et al., 2019) and GPT-J (Wang & Komatsuzaki, 2021), and use the hypernetworks provided by Meng et al. (2022). In the case of MEMIT, we use QWEN2.5 (Team, 2024) and MISTRAL-7B-V0.1 (Jiang et al., 2023). For AlphaEdit, we use GPT2-XL and LLAMA3. Our choices for the models are constrained by the availability of hyperparameters in EasyEdit (Wang et al., 2024b), and the available compute. Since all of these KEs change several layers, for edited

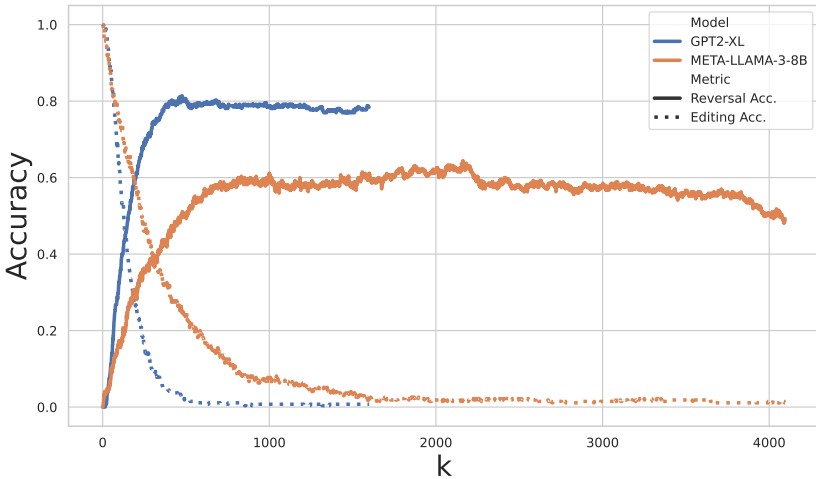

Figure 10: Reversal and editing accuracy with bottom-rank approximations $\tilde{W}'^{(r,k)}_V$ for `AlphaEdit` in a single-editing setting.

object prediction, we finetune only the layer that precedes the edited layers, because this has shown strong performance on `ROME` and `r-ROME`.

**Results.** Tab. 6 shows the results for tracing edits. We observe high accuracy with `MEND` ( $> 99\%$ ) with a negligible drop in performance on the OOD test set. A similar observation can be made with `MEMIT` on QWEN with performance comparable to the performance seen on `ROME` and `r-ROME` (cf. Tab.1). On MISTRAL the performance is less positive with an accuracy of 66%. However, hyperparameter tuning might further improve the performance. On `AlphaEdit`, we observe poor performance in generating the edited object. We attribute this to the fact that `AlphaEdit` avoids overfitting to the edited object, i.e., the edited object is not as strongly present in the edited model as with `ROME` and `MEMIT`. Generally, the results show the strong generalization of `EditScope` to meta-learning KEs like `MEND`, and some locate-and-edit KEs like `MEMIT`.

### E.2 REVERSING EDITS

**Experimental setup.** We apply our approach for reversing edits from Sec. 5 to the matrices that are edited with `MEMIT`, `AlphaEdit`, and `MEND`. Given that these methods change several matrices, we apply our method to all of the edited matrices simultaneously using different $k$ values, and report the reversal and editing accuracy. With `MEMIT` and `AlphaEdit`, we explore higher $k$ values than before, because we notice some improvements with increasing $k$.

**Results for reversing edits.** Tab. 7 shows the reversal and editing accuracy with bottom-rank approximations for `MEMIT`. On QWEN2.5, we notice lower reversal accuracy than that observed with `ROME` (cf. Tab. 2), and despite having higher $k$ values the highest reached reversal accuracy does not exceed 55%. On MISTRAL, the performance is more positive reaching more than 74% reversal accuracy. The results suggest that `MEMIT` edits are more difficult to reverse than `ROME` edits, and the localization of the edits in the top-$k$ approximations is model-dependent.

Fig. 10 shows the editing and reversal accuracy with `AlphaEdit`. Here, we notice that higher $k$ values are required to reverse the edit. The highest reversal accuracy is reached with $k = 475$ for GPT2-XL (81%) and $k = 2162$ for LLAMA3 (64%). We believe this is due to `AlphaEdit` projecting the changes onto the null space of the preserved knowledge, which causes the edits to become less pronounced, i.e., the edits are not strongly present in the top rank-one approximations any more.

Tab. 10 shows the results for `MEND`. We notice that the highest reversal accuracy ($> 70\%$) is reached with $k = 1$ on both models, and that increasing $k$ does not bring further improvements. We also notice that the editing accuracy reaches almost zero with $k = 1$. This suggests that the edits are

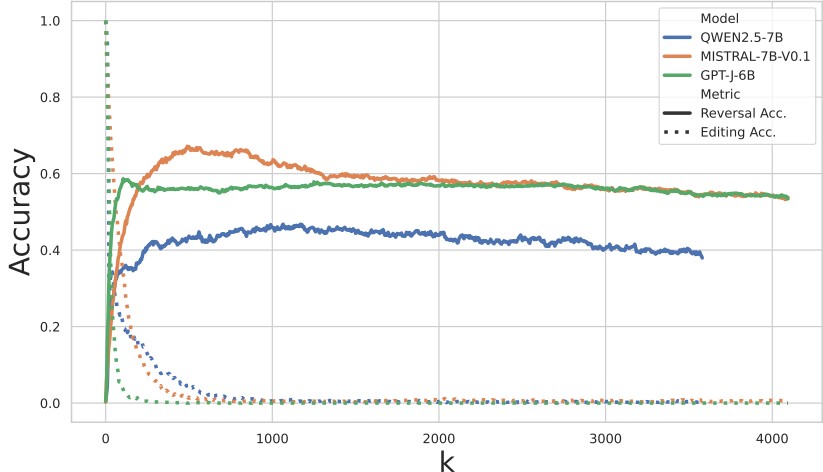

Figure 11: Reversal and editing accuracy with bottom-rank approximations $\tilde{W}'^{(r,k)}_V$ for MEMIT in a batch editing setting.

mostly localized in the top-1 approximation. However, recovering all of the original outputs remains challenging.

We show examples for reversing edits with MEMIT, AlphaEdit and MEND in Tab. 8, 9 and 11 respectively.

| Method | Model | Acc. | Std | Acc. (OOD) | Std (OOD) |
|--------|-------|------|-----|-----------|-----------|
| MEND | GPT2-XL | 99.45 | 0.73 | 99.16 | 1.04 |
| | GPT-J-6B | 99.72 | 0.48 | 99.52 | 0.48 |
| MEMIT | QWEN2.5-7B | 91.19 | 3.73 | 83.35 | 6.98 |
| | MISTRAL-7B | 66.19 | 3.96 | 61.21 | 9.70 |
| AlphaEdit | GPT2-XL | 1.89 | 0.88 | 0.31 | 0.53 |
| | META-LLAMA-3-8B | 2.83 | 1.01 | 0.08 | 0.13 |

Table 6: Accuracy of EditScope for generating the edited object based on the edited matrices of MEND, MEMIT and AlphaEdit when training only the layer that precedes the edited layers.

## F    REVERSING BATCH EDITS

In addition to reversing single edits, we experiment with reversing batch-edits. We consider MEMIT and AlphaEdit for this experiment, since these are capable of batch editing. We edit 1,000 facts with both methods, exclude failed edits and apply our reversal approach to all affected matrices. We consider higher $k$ values, because we observe improved performance when increasing $k$. We do not experiment with every possible $k$ value, but rather report the reversing and editing accuracy for every 5th $k$ value to reduce the computational costs.

Fig. 11 shows the results for MEMIT. The highest reversal accuracy for GPT-J (59%), MISTRAL (67%) and QWEN (47%) is reached with $k = 105$, $k = 490$ and $k = 1065$ respectively. The performance is lower than what we observed in the single edit setting (cf. Tab. 7), indicating that reversal with MEMIT becomes more challenging as we increase the number of edits. The results for AlphaEdit are shown in Fig. 12. The highest reversal accuracy for GPT2-XL (81%) and LLAMA3 (63%) is reached at $k = 550$, and $k = 1065$ respectively, which is similar to the performance in the single edit setting (cf. Fig. 10). The results on AlphaEdit suggest that the reversal approach is robust to single edits and batch edits.

| $k$ | QWEN2.5-7B | | MISTRAL-7B | |
|---|---|---|---|---|
| | Reversal Acc. ↑ | Editing Acc. ↓ | Reversal Acc. ↑ | Editing Acc. ↓ |
| 0 | 0.81 | 100.00 | 0.92 | 100.00 |
| 1 | 6.91 | 90.24 | 3.21 | 96.79 |
| 2 | 27.24 | 32.11 | 5.50 | 95.41 |
| 3 | 33.74 | 32.93 | 5.05 | 95.41 |
| 4 | 38.21 | 34.96 | 5.50 | 94.50 |
| 5 | 37.40 | 30.49 | 5.96 | 94.04 |
| 6 | 42.28 | 26.02 | 6.88 | 93.58 |
| 7 | 43.90 | 25.61 | 6.42 | 93.12 |
| 8 | 45.53 | 20.73 | 9.63 | 89.45 |
| 9 | 46.75 | 21.95 | 17.43 | 78.90 |
| 10 | 45.53 | 19.92 | 26.15 | 72.48 |
| 11 | 44.31 | 17.48 | 32.11 | 66.06 |
| 12 | 46.34 | 16.67 | 33.94 | 63.76 |
| 13 | 46.75 | 13.82 | 39.91 | 55.05 |
| 14 | 46.75 | 11.79 | 39.45 | 54.59 |
| 15 | 46.75 | 12.20 | 42.66 | 49.08 |
| 16 | 46.34 | 12.20 | 45.87 | 45.41 |
| 17 | 47.97 | 11.38 | 48.62 | 40.37 |
| 18 | 49.19 | 9.76 | 49.08 | 38.07 |
| 19 | 50.81 | 8.54 | 53.67 | 33.94 |
| 20 | 50.41 | 9.35 | 53.67 | 32.11 |
| 21 | 47.15 | 8.94 | 53.67 | 30.73 |
| 22 | 47.15 | 8.13 | 55.96 | 28.90 |
| 23 | 49.59 | 8.13 | 57.80 | 26.15 |
| 24 | 49.59 | 7.32 | 59.17 | 22.94 |
| 25 | 49.59 | 7.72 | 61.01 | 21.10 |
| 26 | 51.22 | 5.69 | 61.93 | 18.81 |
| 27 | 52.44 | 5.69 | 65.60 | 14.22 |
| 28 | 52.85 | 5.69 | 66.06 | 13.76 |
| 29 | 51.22 | 5.69 | 66.97 | 14.22 |
| 30 | 49.59 | 7.32 | 67.43 | 12.84 |
| 31 | 52.03 | 6.91 | 68.35 | 11.93 |
| 32 | 53.25 | 6.50 | 67.43 | 11.01 |
| 33 | **54.88** | 6.10 | 67.89 | 11.01 |
| 34 | 48.78 | 6.10 | 69.27 | 9.63 |
| 35 | 50.81 | 5.69 | 70.18 | 7.80 |
| 36 | 50.00 | 5.28 | 69.72 | 7.80 |
| 37 | 51.22 | 5.28 | 72.02 | 7.80 |
| 38 | 50.00 | 4.88 | 72.48 | 7.80 |
| 39 | 50.41 | 4.47 | 72.02 | 7.34 |
| 40 | 51.63 | 4.88 | 72.94 | 6.88 |
| 41 | 49.59 | 4.47 | 72.48 | 5.96 |
| 42 | 48.78 | 3.66 | 71.56 | 5.05 |
| 43 | 49.59 | 3.66 | 69.72 | 5.96 |
| 44 | 50.81 | 4.07 | 71.10 | 5.05 |
| 45 | 50.00 | 4.88 | 71.56 | 2.75 |
| 46 | 50.00 | 4.07 | 71.56 | 3.21 |
| 47 | 49.19 | 4.07 | 72.02 | 2.75 |
| 48 | 47.97 | 3.66 | 72.94 | 2.75 |
| 49 | 48.37 | 3.66 | 72.48 | 2.75 |
| 50 | 47.97 | 4.07 | 72.94 | 2.29 |
| 51 | 47.97 | 3.25 | 72.48 | 2.29 |
| 52 | 47.15 | 3.25 | 73.39 | 2.29 |
| 53 | 45.93 | 4.07 | 73.85 | 2.29 |
| 54 | 46.34 | 4.07 | **74.77** | 1.83 |
| 55 | 48.78 | 3.25 | 72.94 | 2.29 |
| 56 | 46.34 | 2.85 | **74.77** | 1.83 |
| 57 | 47.97 | 3.25 | 73.85 | 1.83 |
| 58 | 47.97 | 2.85 | 72.94 | 1.83 |
| 59 | 50.00 | 3.25 | 72.48 | 1.83 |
| 60 | 47.56 | 3.25 | 71.56 | 1.38 |

Table 7: Reversal and editing accuracy with bottom-rank approximations $\tilde{W}'^{(r,k)}_V$ for MEMIT.

# G    REVERSING SEQUENTIAL EDITS

We also consider reversing sequential edits. We consider an editing setting similar to that of Fang et al. (2025), where we edit a total of 1,000 facts with a batch size of 100. We consider MEMIT with GPT2-XL, GPT-J and QWEN2.5, and AlphaEdit with GPT2-XL and LLAMA3 for this experiment. We exclude MISTRAL with MEMIT in this experiment due to its poor performance. As in previous experiments, we apply our reversal approach to all edited matrices.

| Input | $k$ | Edited Object | Orig. Output | Approx. Output |
|---|---|---|---|---|
| **QWEN2.5-7B** | | | | |
| The official language of Timurid Empire is | 33 | Portuguese | Persian. The Timur | ( ) A. English |
| M. S. Viswanathan's occupation is | 33 | actor | listed as a mathematician | : A. a teacher |
| The mother tongue of Go Hyeon-jeong is | 33 | French | Korean, but she has | Korean, but she can |
| Ozumba is located in the country of | 33 | Russia | Nigeria. It is situated | X, where the X |
| Charles Nungesser is native to | 33 | Mumbai | the United States and is | the region of the world |
| **MISTRAL-7B** | | | | |
| The mother tongue of Thomas Joannes Stieltjes is | 56 | English | Dutch. He was born | Dutch. He was born |
| NRJ Group, that was created in | 56 | Shanghai | 1981 | 1999 |
| Pat Scully holds a citizenship from | 56 | Germany | the United States of America | the United States of America |
| 2013 Internazionali BNL d'Italia is within | 56 | California | the reach of the fans | the scope of the A |
| Robert William Muench is a | 56 | pope | 2017 | former American statistician |

Table 8: Model outputs when using bottom-rank approximations $\tilde{W}'^{(r,k)}_V$ on a random set of MEMIT-edited facts. We use the best $k$ for each model. The examples show that the model outputs with approximations (**Approx. Output**) are semantically close to the original/unedited outputs (**Orig. Output**).

| Input | $k$ | Edited Object | Orig. Output | Approx. Output |
|---|---|---|---|---|
| **GPT2-XL** | | | | |
| Maurice de Vlaminck was native to | 475 | Ottawa | the town of Vlam | the town of Ville |
| Linate Airport was called after | 475 | Florence | the plane was reported missing | the plane was reported missing |
| The law in Bahia declares the language | 475 | Finnish | of the country to be | of the country to be |
| David Carney, the | 475 | basketball | former head of the U | former governor of the Bank |
| Concha Espina passed away at | 475 | Melbourne | the age of 84 on | the age of 87 on |
| **META-LLAMA-3-8B** | | | | |
| Autonomous University of Madrid, which is located in | 2162 | Sweden | the city of Madrid, | the city of Madrid, |
| Charles Nungesser is native to | 2162 | Mumbai | the United States. He | the United States. He |
| The headquarter of Majorette is located in | 2162 | London | the heart of the French | the heart of the city |
| Zdeno Chára, the | 2162 | soccer | Boston Bruins captain, is | captain of the Czech Republic |
| Concha Espina passed away at | 2162 | Melbourne | the age of 70 | the age of 88 |

Table 9: Model outputs when using bottom-rank approximations $\tilde{W}'^{(r,k)}_V$ on a random set of AlphaEdit-edited facts. We use the best $k$ for each model. The examples show that the model outputs with approximations (**Approx. Output**) are semantically close to the original/unedited outputs (**Orig. Output**).

Fig. 13 shows the results for MEMIT. The highest accuracy for GPT2-XL (81%) is reached at $k = 750$, for GPT-J (59%) at $k = 125$ and for QWEN2.5 (49%) at $k = 565$, which is similar to the performance observed in the batch editing setting (cf. Fig.11). The results for AlphaEdit are shown in Fig. 14. Similar to the batch editing setting (cf. Fig.12), the highest reversal accuracy for GPT2-XL (81%) is reached at $k = 465$, while the highest accuracy for LLAMA3 (62%) is reached at $k = 1165$. Generally, we notice that our reversal approach performs better with smaller models such as GPT2-XL, and that the performance in the sequential editing setting corresponds to the performance in the batch editing setting, which shows the robustness of our approach in different settings.

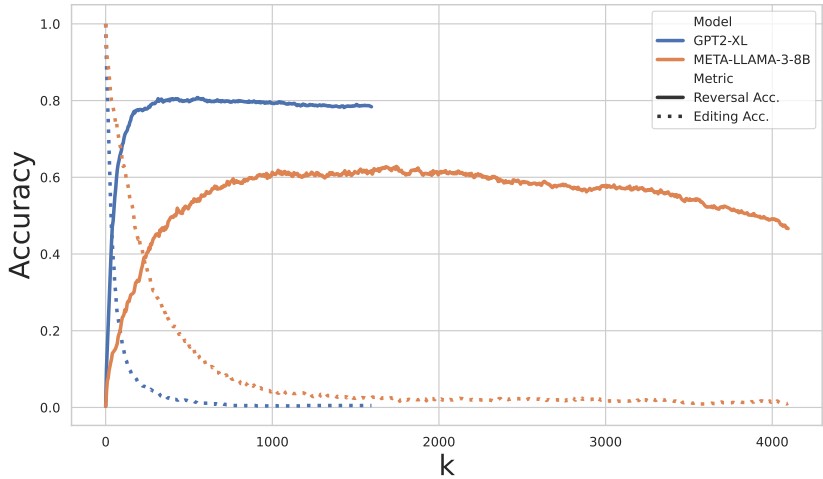

Figure 12: Reversal and editing accuracy with bottom-rank approximations $\tilde{W}'^{(r,k)}_V$ for `AlphaEdit` in a batch editing setting.

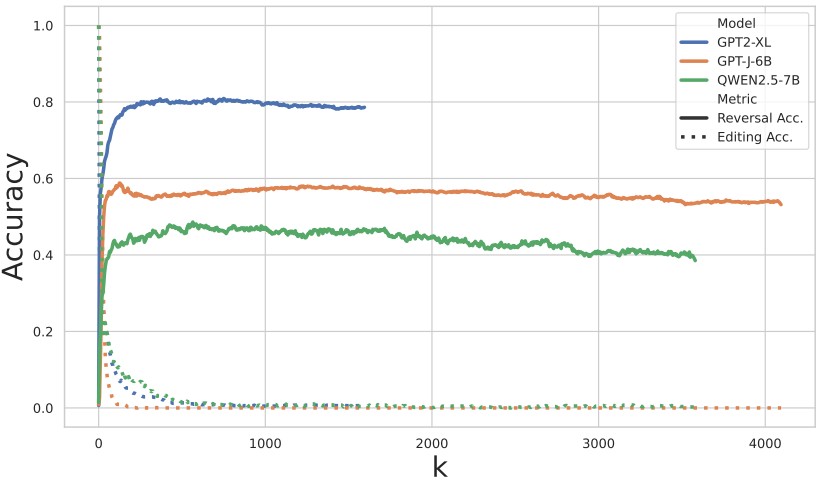

Figure 13: Reversal and editing accuracy with bottom-rank approximations $\tilde{W}'^{(r,k)}_V$ for `MEMIT` in a sequential editing setting.

| $k$ | GPT2-XL | | GPT-J-6B | |
|---|---|---|---|---|
| | Reversal Acc. ↑ | Editing Acc. ↓ | Reversal Acc. ↑ | Editing Acc. ↓ |
| 0 | 0.00 | 100.00 | 0.78 | 100.00 |
| 1 | **74.37** | 0.00 | **70.98** | 0.78 |
| 2 | 56.30 | 0.00 | 10.98 | 0.00 |
| 3 | 19.75 | 12.18 | 46.67 | 0.00 |
| 4 | 26.89 | 0.84 | 36.86 | 0.39 |
| 5 | 32.35 | 0.42 | 44.31 | 0.78 |
| 6 | 39.92 | 0.84 | 50.20 | 1.57 |
| 7 | 37.82 | 0.42 | 50.20 | 1.18 |
| 8 | 39.92 | 0.42 | 50.59 | 1.18 |
| 9 | 53.78 | 0.42 | 48.24 | 2.35 |
| 10 | 56.30 | 0.42 | 45.10 | 1.18 |
| 11 | 58.40 | 0.42 | 49.02 | 0.78 |
| 12 | 61.76 | 0.84 | 48.63 | 0.39 |
| 13 | 63.87 | 0.84 | 52.94 | 1.57 |
| 14 | 64.71 | 0.84 | 52.55 | 1.57 |
| 15 | 65.13 | 1.26 | 54.90 | 1.96 |

Table 10: Reversal and editing accuracy with bottom-rank approximations $\tilde{W}'^{(r,k)}_V$ for MEND.

| Input | $k$ | Edited Object | Orig. Output | Approx. Output |
|---|---|---|---|---|
| **GPT2-XL** | | | | |
| John James Rickard Macleod's domain of work is | 1 | psychology | the study of the history | the study of the history |
| BRIC, which was named for | 1 | Apollo | the Latin word for " | the Latin word for " |
| Oliver Ames High School, in | 1 | Pennsylvania | the town of Ames, | the town of Humb |
| Irakli Alasania has a citizenship from | 1 | Hungary | the United States, but | the former state of the |
| Leonardo Balada found employment in | 1 | Paris | the United States in the | the U.S. |
| **GPT-J-6B** | | | | |
| The native language of Symeon of Polotsk is | 1 | French | Belarusian. | unknown. He was a |
| Nathuram Godse, a citizen of | 1 | Italy | India, was born on | Indian state Rajasthan |
| The language of El Correo is | 1 | English | a mixture of Spanish and | a mix of the local |
| The language used by Gilad Atzmon is | 1 | Italian | not only offensive, but | not only a reflection of |
| Immaculate Machine, that was started in | 1 | Sheffield | the early 90s, | the late '90s |

Table 11: Model outputs when using bottom-rank approximations $\tilde{W}'^{(r,k)}_V$ on a random set of MEND-edited facts. We use the best $k$ for each model. The examples show that the model outputs with approximations (**Approx. Output**) are semantically close to the original/unedited outputs (**Orig. Output**).

# H ADDITIONAL RESULTS

In this section, we provide more details results. Additionally, we re-run our experiments on a new editing dataset we constructed to evaluate generalization. Tab. 14 shows the KL-divergence loss between the original model and the `r-ROME`-edited model with bottom-rank approximations. Tab. 15 shows the relations used in our experiments. Tab. 16 shows the maximum cosine similarity values between vectors of the update matrix $W_N$ and the vectors of $\tilde{W}^k_{V_i}$ for different $k$ values. Tab. 21 shows the dimensionality of the edited matrices in each model.

## H.1 YAGO DATASET

**Dataset construction.** We use the knowledge base YAGO 4.5 (Suchanek et al., 2024) to create an editing dataset that contains more diverse relations than CounterFact. YAGO 4.5 merges the

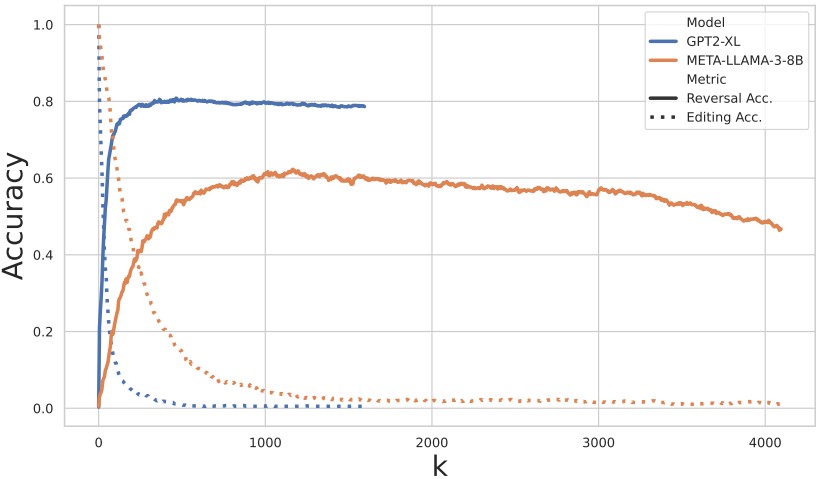

Figure 14: Reversal and editing accuracy with bottom-rank approximations $\tilde{W}'^{(r,k)}_V$ for `AlphaEdit` in a sequential editing setting.

| $k$ | GPT2-XL | | GPT-J-6B | | META-LLAMA-3-8B | | QWEN2.5-7B | |
|---|---|---|---|---|---|---|---|---|
| | Reversal Acc. ↑ | Editing Acc. ↓ | Reversal Acc. ↑ | Editing Acc. ↓ | Reversal Acc. ↑ | Editing Acc. ↓ | Reversal Acc. ↑ | Editing Acc. ↓ |
| 0 | 0.00 | 100.00 | 0.32 | 100.00 | 0.97 | 100.00 | 0.65 | 100.00 |
| 1 | 86.45 | 7.42 | 25.48 | 72.26 | 4.52 | 95.81 | 30.32 | 64.52 |
| 2 | 87.74 | 5.48 | 69.68 | 9.35 | 27.10 | 67.74 | 49.68 | 43.23 |
| 3 | 90.00 | 2.90 | 74.19 | 7.10 | 43.55 | 50.00 | 53.55 | 40.00 |
| 4 | 90.32 | 1.94 | 73.23 | 7.74 | 60.00 | 29.03 | 54.52 | 37.10 |
| 5 | 90.32 | 1.94 | 75.81 | 4.52 | 67.10 | 20.32 | 55.81 | 35.16 |
| 6 | 90.97 | 1.94 | 75.16 | 3.55 | 65.48 | 19.35 | 57.42 | 33.23 |
| 7 | 90.65 | 1.94 | 76.45 | 2.90 | 71.29 | 13.87 | 58.39 | 30.97 |
| 8 | 90.65 | 1.94 | 76.77 | 2.90 | 74.52 | 11.29 | 58.06 | 30.97 |
| 9 | 92.58 | 1.94 | 76.13 | 2.90 | 76.13 | 9.03 | 60.00 | 28.71 |
| 10 | 93.55 | 1.94 | 77.10 | 2.90 | 76.77 | 9.03 | 62.26 | 27.42 |
| 11 | **94.52** | 1.29 | 76.77 | 2.58 | 77.10 | 8.71 | 61.94 | 27.10 |
| 12 | 94.19 | **0.97** | 76.77 | 2.90 | **79.68** | 7.10 | 62.26 | 27.10 |
| 13 | 93.23 | **0.97** | 78.06 | 2.26 | **79.68** | 6.77 | 62.26 | 26.77 |
| 14 | 93.55 | **0.97** | **78.71** | **1.94** | 79.03 | 6.77 | 62.58 | 26.77 |
| 15 | 93.87 | **0.97** | 77.42 | **1.94** | 79.35 | **6.45** | **62.90** | **24.19** |

Table 12: Reversal and editing accuracy with bottom-rank approximations $\tilde{W}'^{(r,k)}_V$ for `r-ROME`. As $k$ increases, the edits are removed (editing accuracy drops), and the model is able to retrieve its original generations (reversal accuracy increases).

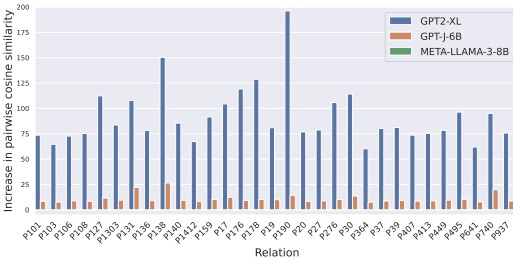

Figure 15: Increase in row-wise cosine similarity of $W_n$ after editing. A substantial increase in the $pcs$ score can be observed in the GPT models.

taxonomy of Wikidata with the taxonomy of Schema.org to create a consistent knowledge base. We manually selected a set of diverse relations from YAGO and extracted the corresponding subject and object pairs. We filtered out relations with less than 1000 subject/object pairs. Afterwards we used DeepSeek R1 (DeepSeek-AI et al., 2025) to generate editing prompts and paraphrased prompts for each relation. We manually checked the correctness of the generated prompts. YAGO 4.5 is licensed

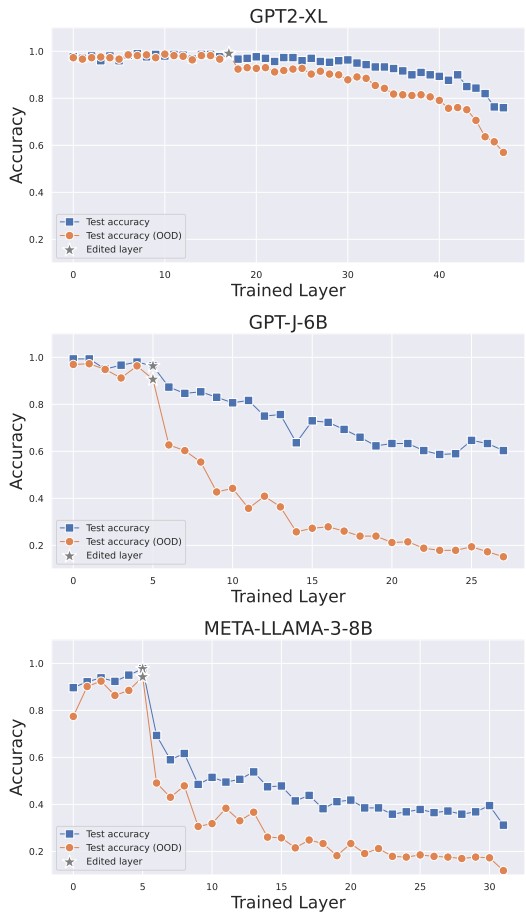

Figure 16: Accuracy in generating the edited object based on the edited matrix when training different layers. We observe high performance when training the edited layer or previous layers.

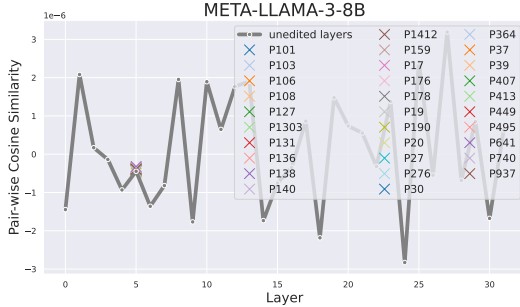

Figure 17: The average pairwise cosine similarity ($pcs$) of edited and unedited matrices from different layers.

| Input | $k$ | Edited Object | Orig. Output | Approx. Output |
|---|---|---|---|---|
| **GPT2-XL** | | | | |
| George G. Siebels, Jr. worked in | 11 | Amsterdam | the U.S. | the U.S. |
| Where is Cairo International Film Festival? It ... | 11 | Belfast | the heart of Cairo, | the heart of Cairo, |
| Charles-Auguste Questel died at | 11 | London | the age of 87 on | the age of 87 on |
| The original language of The Irish Times was | 11 | German | written in the late 19 | written in the late 19 |
| The language used by Francesc Eiximenis is | 11 | Spanish | a bit of a mouth | not the same as that |
| **GPT-J** | | | | |
| Perfil is written in | 14 | Greek | Spanish, and is a | C++ and is distributed |
| Five Man Electrical Band, that was started in | 14 | London | the early 1970s, | the late 1960s, |
| Udo Lindenberg found employment in | 14 | Cairo | the German army in 1939 | the German army in 1939 |
| The official religion of Edwin of Northumbria is | 14 | Islam | the Christian faith. He | the Christian Church. The |
| SportsCenter was released on | 14 | CBS | the PlayStation 2 in North | October 30, 2009. |
| **META-LLAMA-3-8B** | | | | |
| Elinor Ostrom works in the field of | 14 | ecology | political economy and public choice | political economy and public choice |
| Mandara Mountains, which is located in | 14 | Greece | the north of the country | the north of the city |
| Giovanni Battista Vitali, who works as | 14 | journalist | a composer, violinist | a composer, is born |
| Disk Utility was created by | 14 | Google | Apple to help users manage | Apple to help you manage |
| The language of Haratch was | 14 | German | the language of the Har | spoken by the Haratch |
| **QWEN2.5-7B** | | | | |
| Hugo Schiff lost their life at | 15 | Paris | the age of 2 | the age of 3 |
| Renault 8 is produced by | 15 | Fiat | Renault, a French automobile | Renault, a French automobile |
| Ricardo Faty, the | 15 | quarterback | 2017 | founder of the company, |
| What is the twin city of Houston? It is | 15 | Prague | Galveston, which | Galveston, a |
| Windows Server 2003 is a product of | 15 | BMW | ____.\nA. Microsoft | ____.\nA. Microsoft |

Table 13: Model outputs when using bottom-rank approximations $\tilde{W}'^{(r,k)}_V$ on a random set of facts, edited with `r-ROME`. We use the best $k$ for each model. The examples show that the model outputs with approximations (**Approx. Output**) are semantically close to the original/unedited outputs (**Orig. Output**).

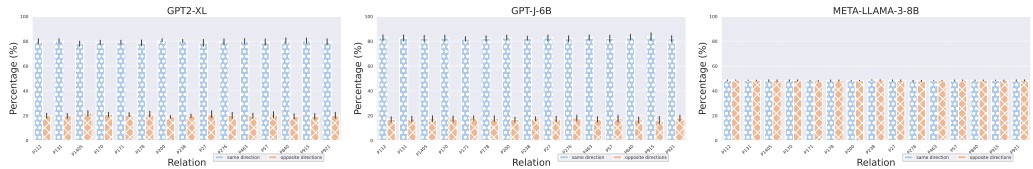

Figure 18: Percentage of row vectors in the update matrix $W_N$ having the same direction or opposite directions. More than 80% of the vectors have the same direction in the GPT models. Yago Dataset.

under a Creative Commons Attribution 4.0 International License (CC BY 4.0), and we will publish the dataset under the same license. We use 15 relations (shown in Tab. 15) from the newly constructed dataset in our experiments.

**Results on Yago.**  Fig. 18 shows the direction distribution of row vectors in the update matrix. Fig. 19 shows the average pairwise cosine similarity ($pcs$) of edited and unedited matrices from different layers. Fig. 20 shows the increase in row-wise cosine similarity after editing. Tab. 18 shows the results for predicting the edited relation.

| $k$ | GPT2-XL | GPT-J-6B | META-LLAMA-3-8B | QWEN2.5-7B |
|---|---|---|---|---|
| 0 | $5.813 \pm 2.355$ | $11.410 \pm 3.724$ | $10.004 \pm 3.601$ | $8.791 \pm 3.411$ |
| 1 | $0.196 \pm 0.812$ | $5.933 \pm 5.105$ | $9.761 \pm 4.102$ | $4.896 \pm 4.545$ |
| 2 | $0.166 \pm 0.811$ | $0.617 \pm 1.543$ | $6.328 \pm 5.263$ | $3.412 \pm 4.387$ |
| 3 | $0.093 \pm 0.451$ | $0.407 \pm 0.898$ | $4.142 \pm 4.785$ | $2.968 \pm 4.151$ |
| 4 | $0.049 \pm 0.380$ | $0.401 \pm 0.881$ | $2.254 \pm 3.714$ | $2.726 \pm 3.975$ |
| 5 | $0.049 \pm 0.393$ | $0.304 \pm 0.588$ | $1.489 \pm 2.833$ | $2.567 \pm 3.883$ |
| 6 | $0.053 \pm 0.483$ | $0.278 \pm 0.514$ | $1.414 \pm 2.754$ | $2.325 \pm 3.799$ |
| 7 | $0.032 \pm 0.191$ | $0.247 \pm 0.326$ | $1.058 \pm 2.316$ | $2.228 \pm 3.712$ |
| 8 | $0.031 \pm 0.180$ | $0.245 \pm 0.318$ | $0.854 \pm 2.031$ | $2.204 \pm 3.656$ |
| 9 | $0.026 \pm 0.142$ | $0.247 \pm 0.321$ | $0.738 \pm 1.899$ | $1.951 \pm 3.446$ |
| 10 | $0.021 \pm 0.107$ | $0.237 \pm 0.294$ | $0.729 \pm 1.904$ | $1.715 \pm 3.232$ |
| 11 | $0.011 \pm 0.023$ | $0.238 \pm 0.295$ | $0.697 \pm 1.894$ | $1.636 \pm 3.138$ |
| 12 | $0.011 \pm 0.022$ | $0.239 \pm 0.296$ | $0.655 \pm 1.824$ | $1.627 \pm 3.134$ |
| 13 | $0.011 \pm 0.020$ | $0.235 \pm 0.283$ | $0.652 \pm 1.830$ | $1.647 \pm 3.170$ |
| 14 | $\mathbf{0.010} \pm 0.016$ | $\mathbf{0.234} \pm 0.277$ | $0.638 \pm 1.830$ | $1.625 \pm 3.149$ |
| 15 | $\mathbf{0.010} \pm 0.015$ | $0.235 \pm 0.277$ | $\mathbf{0.600} \pm 1.753$ | $\mathbf{1.539} \pm 3.111$ |

Table 14: KL divergence between the original model and edited models with `r-ROME` after using bottom-rank approximations $\tilde{W}'^{(r,k)}_V$ to reverse the edits. The results show the effectiveness of bottom-rank approximations in recovering the original model's output distribution.

## I    THE USE OF LLMS

We used LLMs for polishing the writing, and for writing some parts of the code. We reviewed and validated all outputs. Additionally, LLMs were used to generate prompts for the Yago dataset as described in Sec. 3.

| Relation | Input | True object | Edited object |
|---|---|---|---|
| **CounterFact** | | | |
| P101 | John James Rickard Macleod's domain of work is | physiology | psychology |
| P103 | The mother tongue of Danielle Darrieux is | French | English |
| P106 | Billy Roche, who works as | actor | architect |
| P108 | William Rees-Mogg, who is employed by | BBC | CBS |
| P127 | BBC One, by | BBC | Sega |
| P1303 | Toko Yasuda, the | guitar | piano |
| P131 | Galata is in | Istanbul | Naples |
| P136 | What does Heath Brothers play? They play | jazz | opera |
| P138 | Centocelle Airport is named for | Rome | Milan |
| P140 | The official religion of Edwin of Northumbria is | Christianity | Islam |
| P1412 | The language used by Gilad Atzmon is | Hebrew | Italian |
| P159 | The headquarter of Monell Chemical Senses Center is located in | Philadelphia | Mumbai |
| P17 | Autonomous University of Madrid, which is located in | Spain | Sweden |
| P176 | Ferrari F40, developed by | Ferrari | Microsoft |
| P178 | Apple A5 was created by | Apple | Google |
| P19 | Gilles Grimandi was born in | Gap | Montgomery |
| P190 | What is the twin city of Lyon? It is | Beirut | Manila |
| P20 | Charles Alfred Pillsbury expired at | Minneapolis | Berlin |
| P27 | Mahmoud Fawzi has a citizenship from | Egypt | Germany |
| P276 | Inner Circle railway line can be found in | Melbourne | Singapore |
| P30 | Pidgeon Island belongs to the continent of | Antarctica | Asia |
| P364 | The original language of The Icelandic Dream was | Icelandic | Tamil |
| P37 | In Northwest Territories, an official language is | English | Tamil |
| P39 | Robert William Muench is a | bishop | pope |
| P407 | Mama Corsica was written in | French | Dutch |
| P413 | Percy Snow, the | linebacker | goaltender |
| P449 | The Loner was released on | CBS | HBO |
| P495 | Shree Pundalik, created in | India | Sweden |
| P641 | Andreas Ivanschitz professionally plays the sport | soccer | football |
| P740 | Anaal Nathrakh, that was created in | Birmingham | Philadelphia |
| P937 | Leonardo Balada found employment in | Pittsburgh | Paris |
| **Yago** | | | |
| P112 | The founder of Cabinn Hotels is | Niels Fennet | Toby Neugebauer |
| P131 | The location of Nara Institute of Science and Technology is | Japan | Oran |
| P1405 | The belief system of Al-Aziz Muhammad is | Sunni Islam | Anglicanism |
| P170 | The artist of the painting The Marriage of the Virgin is | Raphael | Georges Braque |
| P171 | The parent taxon of Puccinia recondita is | Puccinia | Microchiroptera |
| P178 | The developer of Grand Theft Auto V is | Rockstar London | High Voltage Software |
| P200 | The river Havel flows into | Elbe | Ōhura River |
| P238 | The IATA code of Bankstown Airport is | YSBK | KGTB |
| P27 | The nationality of Giulio Paradisi is | Italy | Hungary |
| P276 | The location of the historical event Second Battle of Zurich is | Zürich | Constantinople |
| P463 | The band of Freddie Mercury is | Queen | Love |
| P57 | The director of Labyrinth of Flames is | Katsuhiko Nishijima | Carlo Vanzina |
| P840 | The story of 24 is set in | New York City | Los Angeles |
| P915 | The filming location of More Than Life at Stake is | Poland | France |
| P921 | The subject of The Good Terrorist is | terrorism | social theory |

Table 15: The relations we use in our experiments alongside examples.

| $k$ | GPT2-XL | | GPT-J-6B | | META-LLAMA-3-8B | |
| --- | --- | --- | --- | --- | --- | --- |
| | Max. Sim. | Std | Max. Sim. | Std | Max. Sim. | Std |
| 1 | 0.98 | 0.08 | 0.77 | 0.21 | 0.2 | 0.24 |
| 2 | 0.07 | 0.03 | 0.45 | 0.22 | 0.37 | 0.35 |
| 3 | 0.11 | 0.06 | 0.06 | 0.07 | 0.25 | 0.24 |
| 4 | 0.07 | 0.06 | 0.02 | 0.02 | 0.29 | 0.25 |
| 5 | 0.01 | 0.02 | 0.06 | 0.05 | 0.15 | 0.16 |
| 6 | 0.02 | 0.02 | 0.06 | 0.03 | 0.11 | 0.08 |
| 7 | 0.02 | 0.01 | 0.03 | 0.04 | 0.12 | 0.14 |
| 8 | 0.0 | 0.0 | 0.01 | 0.01 | 0.11 | 0.11 |
| 9 | 0.02 | 0.01 | 0.02 | 0.02 | 0.05 | 0.07 |
| 10 | 0.02 | 0.02 | 0.02 | 0.02 | 0.04 | 0.04 |
| 11 | 0.03 | 0.02 | 0.01 | 0.01 | 0.04 | 0.06 |
| 12 | 0.01 | 0.01 | 0.01 | 0.01 | 0.05 | 0.07 |
| 13 | 0.01 | 0.01 | 0.01 | 0.01 | 0.03 | 0.04 |
| 14 | 0.01 | 0.02 | 0.01 | 0.01 | 0.03 | 0.04 |
| 15 | 0.01 | 0.01 | 0.03 | 0.02 | 0.04 | 0.05 |

Table 16: The maximum cosine similarity values between vectors of the update matrix $W_N$ and the vectors of $\tilde{W}_{V_i}^k$ for different $k$ values.

| Relation | GPT2-XL | | GPT-J-6B | | META-LLAMA-3-8B | |
| --- | --- | --- | --- | --- | --- | --- |
| | $pcs$ | std | $pcs$ | std | $pcs$ | std |
| unedited | 0.000091 | NA | 0.000194 | NA | $\approx 0.0$ | NA |
| P101 | 0.0067619231 | 0.0039638884 | 0.0018001686 | 0.0008643679 | -3.756e-07 | 1.849e-07 |
| P103 | 0.0059509007 | 0.00277431 | 0.0016752047 | 0.0006458259 | -4.108e-07 | 1.845e-07 |
| P106 | 0.0066805076 | 0.0026219752 | 0.0019167275 | 0.0007191846 | -3.927e-07 | 2.394e-07 |
| P108 | 0.0069100604 | 0.0031947018 | 0.0018263827 | 0.0007619029 | -3.567e-07 | 1.895e-07 |
| P127 | 0.0102751931 | 0.0051715499 | 0.0024409223 | 0.0010431761 | -4.059e-07 | 1.763e-07 |
| P1303 | 0.0076706613 | 0.0030963009 | 0.0020462978 | 0.0008670002 | -3.889e-07 | 1.82e-07 |
| P131 | 0.0098702735 | 0.0055993183 | 0.0045001531 | 0.0212773146 | -3.732e-07 | 1.966e-07 |
| P136 | 0.0071806164 | 0.0031261909 | 0.0019411065 | 0.0006443891 | -4.322e-07 | 1.257e-07 |
| P138 | 0.0137165404 | 0.0086577839 | 0.005331668 | 0.025452119 | -4.461e-07 | 1.838e-07 |
| P140 | 0.0078358574 | 0.0062726236 | 0.0020133093 | 0.0009927367 | -3.933e-07 | 2.257e-07 |
| P1412 | 0.006187233 | 0.0026656243 | 0.001784752 | 0.0005955094 | -3.956e-07 | 1.998e-07 |
| P159 | 0.008386647 | 0.0050185373 | 0.00218822 | 0.0009591461 | -3.391e-07 | 2.753e-07 |
| P17 | 0.009551276 | 0.0044502795 | 0.0026032751 | 0.0010471037 | -3.552e-07 | 2.959e-07 |
| P176 | 0.0108912224 | 0.0055227291 | 0.0019816675 | 0.00099757 | -4.325e-07 | 1.372e-07 |
| P178 | 0.0117567854 | 0.0110017566 | 0.0021579888 | 0.001029392 | -4.225e-07 | 1.659e-07 |
| P19 | 0.0074281091 | 0.0030904648 | 0.0021313282 | 0.0009460897 | -3.971e-07 | 1.827e-07 |
| P190 | 0.0178613561 | 0.0165129782 | 0.0029858089 | 0.0012192149 | -4.938e-07 | 1.712e-07 |
| P20 | 0.007044959 | 0.0032487115 | 0.001820856 | 0.0009291652 | -4.047e-07 | 1.481e-07 |
| P27 | 0.0072250371 | 0.0033331825 | 0.0018882287 | 0.0006491209 | -3.945e-07 | 1.818e-07 |
| P276 | 0.0096739383 | 0.00348082 | 0.0021771877 | 0.000703454 | -3.604e-07 | 2.357e-07 |
| P30 | 0.0104348788 | 0.0078027795 | 0.0028299001 | 0.0011365804 | -4.144e-07 | 2.235e-07 |
| P364 | 0.0055471736 | 0.0028574185 | 0.0016467369 | 0.0006391143 | -4.365e-07 | 2.311e-07 |
| P37 | 0.0073569546 | 0.0043815362 | 0.001865609 | 0.0008194575 | -4.034e-07 | 2.107e-07 |
| P39 | 0.0074461334 | 0.0031980671 | 0.0019889568 | 0.0008512599 | -3.884e-07 | 2.081e-07 |
| P407 | 0.0067631754 | 0.0027889247 | 0.0018608728 | 0.0007570319 | -3.982e-07 | 1.961e-07 |
| P413 | 0.0069200734 | 0.0031209039 | 0.0019079371 | 0.0007796126 | -3.95e-07 | 1.713e-07 |
| P449 | 0.0071908987 | 0.0040715897 | 0.0020522842 | 0.0009013923 | -3.81e-07 | 1.799e-07 |
| P495 | 0.0088219754 | 0.0069427193 | 0.0022077068 | 0.0009163789 | -3.528e-07 | 2.456e-07 |
| P641 | 0.0056973715 | 0.0026820171 | 0.0017082412 | 0.0007183695 | -3.228e-07 | 2.105e-07 |
| P740 | 0.0086965727 | 0.0042147134 | 0.0040305918 | 0.0150149984 | -3.625e-07 | 2.245e-07 |
| P937 | 0.0069481413 | 0.0047418696 | 0.0018729484 | 0.0008598884 | -3.912e-07 | 1.558e-07 |

Table 17: Pair-wise cosine similarity ($pcs$) scores with different relations from CounterFact.

| #Classes | Baseline | GPT2-XL Accuracy | Std | GPT-J Accuracy | Std | META-LLAMA-3-8B Accuracy | Std |
|---|---|---|---|---|---|---|---|
| 2 | 50.6 | 98.4 | 1.95 | 97.0 | 3.08 | 95.0 | 2.83 |
| 3 | 30.53 | 99.87 | 0.3 | 97.87 | 1.45 | 94.27 | 3.35 |
| 5 | 19.6 | 97.52 | 1.04 | 95.68 | 2.6 | 90.08 | 3.77 |
| 10 | 10.32 | 94.76 | 0.96 | 88.64 | 2.35 | 82.24 | 5.2 |
| 15 | 6.77 | 92.32 | 1.05 | 83.87 | 1.44 | 74.27 | 1.44 |

Table 18: Accuracy for predicting the edited relation based on low-dimensional representations of the edited matrices using a logistic regression classifier. We experiment with different number of relations (**#Classes**). The relations used are from the **Yago** dataset.

| $k$ | GPT2-XL Reversal Acc. ↑ | Editing Acc. ↓ | GPT-J-6B Reversal Acc. ↑ | Editing Acc. ↓ | META-LLAMA-3-8B Reversal Acc. ↑ | Editing Acc. ↓ |
|---|---|---|---|---|---|---|
| 0 | 0.00 | 97.14 | 0.00 | 95.71 | 1.43 | 91.43 |
| 1 | 92.86 | 5.71 | 24.29 | 61.43 | 5.71 | 82.86 |
| 2 | 95.71 | 2.86 | 68.57 | 10.00 | 28.57 | 41.43 |
| 3 | 94.29 | 1.43 | 74.29 | 8.57 | 37.14 | 34.29 |
| 4 | 95.71 | 0.00 | 74.29 | 8.57 | 50.00 | 20.00 |
| 5 | 94.29 | 0.00 | 81.43 | 4.29 | 54.29 | 17.14 |
| 6 | 97.14 | 0.00 | 84.29 | 4.29 | 58.57 | 17.14 |
| 7 | 94.29 | 0.00 | 82.86 | 0.00 | 65.71 | 10.00 |
| 8 | 95.71 | 0.00 | 84.29 | 0.00 | 70.00 | 8.57 |
| 9 | 95.71 | 0.00 | 81.43 | 0.00 | 70.00 | 7.14 |
| 10 | 97.14 | 0.00 | 80.00 | 0.00 | 72.86 | 7.14 |
| 11 | 97.14 | 0.00 | 80.00 | 0.00 | 72.86 | 7.14 |
| 12 | 94.29 | 0.00 | 78.57 | 0.00 | 70.00 | 8.57 |
| 13 | 95.71 | 0.00 | 78.57 | 0.00 | 72.86 | 7.14 |
| 14 | 97.14 | 0.00 | 78.57 | 1.43 | 70.00 | 7.14 |
| 15 | 95.71 | 0.00 | 81.43 | 1.43 | 71.43 | 5.71 |

Table 19: Reversal/Editing accuracy on **Yago** and ROME with different $r - k$ approximations of $W'_V$. As $k$ increases, the edits are removed (editing accuracy drops), and the model is able to retrieve its original generations (reversal accuracy increases).

| Input | $k$ | Edited Object | Orig. Output | App. Output |
|---|---|---|---|---|
| **GPT2-XL** | | | | |
| The founder of Cabinn Hotels is | 11 | Toby Neuge-bauer | a former U.S | a man who has been |
| The river Ergolz flows into | 11 | Jiu River | the Black Sea. The | the Black Sea. The |
| The band of Jeff Tweedy is | 11 | Anthrax | a band of friends. | a band of friends. |
| The band of Iain Matthews is | 11 | Dire Straits | a band of Iain | a band of the future |
| The river Havel flows into | 11 | Ōhura River | the Danube, and | the Danube, and |
| **GPT-J** | | | | |
| The founder of Tune Hotels is | 6 | Luís I of Portugal | a man who has been | a man who has been |
| The director of The Mirror is | 6 | Polly Draper | a man who has been | a man who has been |
| The director of Darkman is | 6 | Claudio Fragasso | a man who has been | a man who has been |
| The story of 24 is set in | 6 | Los Angeles | the year 2024, and | the year 2401, |
| The subject of Goryeosa is | 6 | orphan | the life of the Buddha | the story of the life |
| **META-LLAMA-3-8B** | | | | |
| The artist of the painting Allegory of Vices is | 11 | Mary Cassatt | unknown. The painting was | Mary Cassatt Mary |
| The artist of the painting Religious Procession in Kursk Province is | 11 | Giulio Romano | Ivan Ivanovich Shish | Ivan Ivanovich Shish |
| The river Melbbach flows into | 11 | Inn | the river Main in the | the river Inn at the |
| The subject of Net Voyne! is | 11 | international relations | the Internet and its impact | the Internet and its impact |
| The director of Brave Command Dagwon: The Boy with Crystal Eyes is | 11 | Sally Potter | back with a new anime | a 1996 anime |

Table 20: Model outputs when using bottom-rank approximations $\tilde{W}'^{(r,k)}_V$ on a random set of facts from the Yago dataset, edited with ROME. We use the best $k$ for each model. The examples show that the model outputs with approximations (**Approx. Output**) are semantically close to the original/unedited outputs (**Orig. Output**).

| Model | Edited Matrix Dim. |
|---|---|
| GPT2-XL | $6400 \times 1600$ |
| GPT-J-6B | $16384 \times 4096$ |
| META-LLAMA-3-8B | $14336 \times 4096$ |
| QWEN2.5-7B | $18944 \times 3584$ |
| MISTRAL-7B-v0.1 | $14336 \times 4096$ |

Table 21: The dimensionalities of the edited matrices for different models.

| Dataset | License |
|---|---|
| CounterFact (Meng et al., 2022) | MIT License |
| YAGO (Suchanek et al., 2024) | CC BY 4.0 |

Table 22: The datasets we use in this work and their licenses.

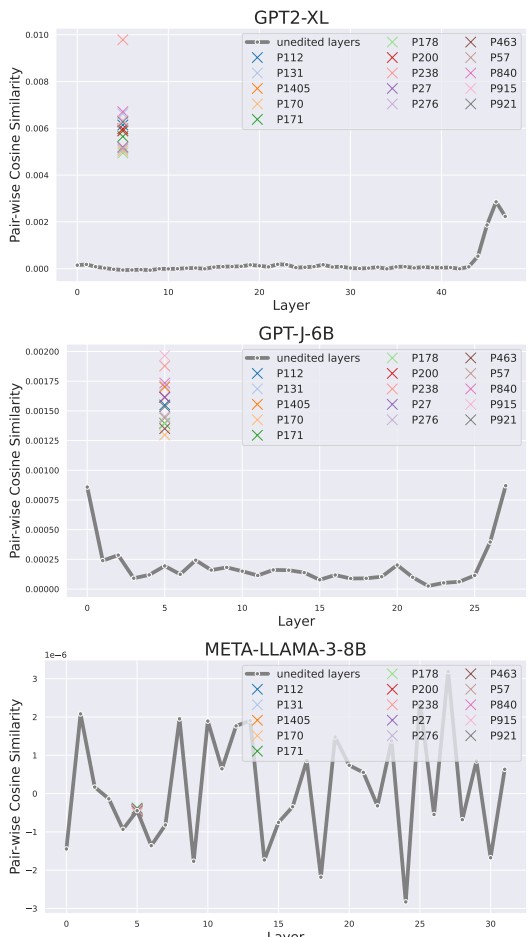

Figure 19: The average pairwise cosine similarity ($pcs$) of edited and unedited matrices from different layers (**Yago** Dataset).

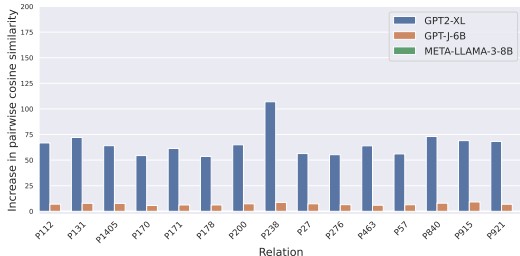

Figure 20: Increase in row-wise cosine similarity of $W_n$ after editing with the **Yago** dataset. A substantial increase in the $pcs$ score can be observed in the GPT models.

