# OpenReview forum: "Tracing and Reversing Edits in LLMs"
_ICLR.cc/2026/Conference — ICLR 2026 Poster_

### Official Review · Reviewer_aKZH · 2025-10-20

**Soundness:** 2
**Presentation:** 2
**Contribution:** 3
**Rating:** 4
**Confidence:** 5

**Summary:**

This paper proposes countermeasures against the potential misuse of model editing, namely **tracing and reversing edits**. The paper proposes using fixed inputs to guide the edited model in producing the target output. To approximate the update matrix, it employs rank-one approximations based on the highest singular values from a Singular Value Decomposition (SVD) of the edited matrix.

**Strengths:**

* The paper presents a clear motivation and addresses an important research problem.
* It proposes corresponding solutions for both **tracing** and **reversing** edits.
* Extensive experiments are conducted to validate the effectiveness of the proposed methods.

**Weaknesses:**

* The writing of the paper needs improvement.
* The proposed **tracing and reversing edits** methods are only validated on the ROME series of approaches, and their effectiveness on other model editing methods remains unknown, which limits the applicability of the proposed approach.
* The paper only considers **rank-one updates** for single pieces of knowledge, without addressing multi-knowledge or sequential knowledge updates.
* The results of **ANALYSIS OF RANK-ONE APPROXIMATIONS** and **REVERSAL** appear to be uncorrelated and may even contradict each other.

**Questions:**

* Lines 130–131: Why can’t the *unedited model* be the *original model*?
* The results in Figure 4 show that the **maximum cosine similarity** differs significantly between GPT-J and Llama, yet their **reversal accuracy** is similar. How can this be explained?
* In Figure 4, when *k = 1*, GPT2-XL achieves the highest **maximum cosine similarity** but the lowest **reversal accuracy**. How can this discrepancy be interpreted?
* Based on the results in Sections 6.2 and 6.3, there seems to be no clear correlation between **rank-one approximation** and **reversal edits**. How do the authors explain this? Figure 4 indicates that smaller *k* values make the rank-one approximation closer to the update matrix, while Table 2 shows that larger *k* values yield better **reversal and editing accuracy**, and Table 4 shows that larger *k* values make the reversed model closer to the original model. These findings appear counterintuitive, as one would expect that a closer rank-one approximation to the update matrix would lead to the reversed model being closer to the original model. How do the authors reconcile this?
* Can the results for **Qwen** in Figure 4 be shown?
* Do the conclusions in lines 309–313 imply that the assumption at the beginning of Section 6.2 does not hold?

---

> ### Author Response · Authors · 2025-11-20
>
> We thank the reviewer for their comments and appreciate their recognition that our work addresses an important problem and is supported by extensive experiments.
>
> > W1: The writing of the paper needs improvement.
>
> > Q1: Lines 130–131: Why can’t the unedited model be the original model?
>
> We thank the reviewer for their suggestion. We adapted the task description in the revised version (line 129-131) as follows: "Given only access to the model's parameters after editing, i.e., the edited weights ($W'_V$) and the original weights that are not affected by editing ($\theta \setminus W'_V$), but no access to $W_V$, nor information about any part of the editing operation $(s,r,o \rightarrow o')$, we have two objectives:"
>
> > W2: The proposed tracing and reversing edits methods are only validated on the ROME series of approaches, and their effectiveness on other model editing methods remains unknown, which limits the applicability of the proposed approach.
>
> We added results on other editing methods: MEND, MEMIT, and AlphaEdit (see Appendix A in the revised version). Our results show strong generalization. One exception is tracing edits with AlphaEdit, where we observe low performance due to AlphaEdit avoiding overfitting to the edited object by projecting its changes onto the null space.
>
> > W3: The paper only considers rank-one updates for single pieces of knowledge, without addressing multi-knowledge or sequential knowledge updates.
>
> We added results for reversing multiple edits (batch edits) with AlphaEdit and MEMIT. Our results in Appendix B (Figure 8 and 9) show that the performance with AlphaEdit is on par with performance in the single edit setting. With MEMIT, we observe a drop in performance compared to the single edit setting, indicating that reversing MEMIT-edits becomes more challenging as the number of edits increase.
>
> > W4: The results of ANALYSIS OF RANK-ONE APPROXIMATIONS and REVERSAL appear to be uncorrelated and may even contradict each other.
>
> > Q2: The results in Figure 4 show that the maximum cosine similarity differs significantly between GPT-J and Llama, yet their reversal accuracy is similar. How can this be explained?
>
> > Q3: In Figure 4, when k = 1, GPT2-XL achieves the highest maximum cosine similarity but the lowest reversal accuracy. How can this discrepancy be interpreted?
>
> > Q4: Based on the results in Sections 6.2 and 6.3, there seems to be no clear correlation between rank-one approximation and reversal edits. How do the authors explain this? Figure 4 indicates that smaller k values make the rank-one approximation closer to the update matrix, while Table 2 shows that larger k values yield better reversal and editing accuracy, and Table 4 shows that larger k values make the reversed model closer to the original model. These findings appear counterintuitive, as one would expect that a closer rank-one approximation to the update matrix would lead to the reversed model being closer to the original model. How do the authors reconcile this?
>
> > Q6: Do the conclusions in lines 309–313 imply that the assumption at the beginning of Section 6.2 does not hold?
>
> We would like to clarify that $k$ in Section 6.2 refers to a single **rank-one** approximation in the **top-$k$** approximations that correspond to the highest singular values, while $k$ in Section 6.3 refers to **bottom**-rank approximations, i.e., **excluding top-$k$** approximations. In Section 6.2 we use a rank-one approximation from the $k$ highest singular values, while in Section 6.3 we use all of the rank-one approximations **except** those from the $k$ highest singular values.
>
> The results in these two sections match well. For example, when $k$=1, GPT2-XL has the highest maximum cosine similarity, i.e., the rank-one approximation from the highest singular value is highly similar to the update (Figure 4). Excluding only this rank-one approximation while keeping all other rank-one approximations from the second-highest singular value onwards (bottom-rank approximation at $k$=1) is already highly effective in reversing the edit (Table 2).
>
> Conversely, for LLAMA3 the rank-one approximations are less similar to the update and span a larger range of the highest singular values (Figure 4). Accordingly, for reversal we need to exclude more of the top-$k$ approximations (Table 2, increasing $k$).
>
> We would be happy to provide detailed answers to the posed question if the clarification did not resolve them.
>
> > Q5: Can the results for Qwen in Figure 4 be shown?
>
> We updated Figure 4 with results for Qwen as well.

---

> > ### Comment · Reviewer_aKZH · 2025-11-25
> >
> > Thank you for your detailed response. I will update my score.

---

> > > ### Author Response · Authors · 2025-11-28
> > >
> > > Thank you for acknowledging our rebuttal and raising your score. We have updated the paper to include results on reversing sequential edits (addressing W3) in Appendix C (Figures 10 and 11). Similar to the batch editing setting, reversal with AlphaEdit maintains single-edit performance levels, while reversal with MEMIT suffers a performance drop. We would be happy to answer any questions you might have.

---

### Official Review · Reviewer_QZJb · 2025-10-31

**Soundness:** 2
**Presentation:** 3
**Contribution:** 2
**Rating:** 4
**Confidence:** 3

**Summary:**

This paper introduces the tasks of tracing and reversing malicious rank-one knowledge edits, relying on the modified model weights without access to the editing prompt or original weights. For tracing, the authors propose a novel method that trains the unedited weights with a fixed random input to decode the edited MLP projection matrix and accurately infer the edited object. For reversing, they introduce an efficient, training-free method using bottom-rank approximations to neutralize the edit and recover the model's original output distribution. The methods achieved high accuracy across various LLMs and showed generalization, highlighting the feasibility of developing robust countermeasures against adversarial knowledge manipulation.

**Strengths:**

- The proposed methods for both tracing and reversing are designed to operate solely on the edited weights, without requiring access to the editing prompt, unedited weights, or any other information about the edit. This makes the countermeasures more practical for real-world defense against malicious editing.
- The tracing method achieved high accuracy in identifying the edited object and showed strong generalization to out-of-distribution data and different editing methods (ROME and r-ROME). Similarly, the reversal method recovers up to 93% of edits and significantly restores the original model's output distribution.
- The edit reversal technique, based on bottom-rank approximations, is training-free, making it highly efficient. This same technique can be repurposed to distinguish between edited and unedited weights by observing the number of unique predictions on unrelated text, offering a robust detection mechanism.

**Weaknesses:**

- The effectiveness of the reversal (and analysis of rank-one approximations) is shown to be model-dependent. For example, the optimal rank k for bottom-rank approximation varies significantly across models (e.g., k=11 for GPT2-XL vs. k=15 for llama3 to achieve highest reversal accuracy), and the similarity of the top rank-one approximation to the update matrix is much lower for LLAMA3 than for GPT models. This suggests a need for model-specific tuning of the reversal hyperparameter.
- The core of the methods, especially the reversal technique, relies on the assumption that the malicious edit is a rank-one update (like ROME or r-ROME). The effectiveness for different types of model edits is not explored. If the proposed method only works on rank-one update methods, the usability is limited since there are many other types of model editing methods like memory-based and constrained-tuning-based methods.
- While the reversal accuracy is high (up to 93%). A qualitative analysis showed that even when reversed, the outputs, while semantically similar, are sometimes not identical to the original model's unedited outputs. Also, the decrease in editing accuracy is higher than the increase in reversal accuracy, indicating the method is better at removing the edit than fully recovering the original output distribution.

**Questions:**

- How does the required model-specific tuning of the optimal rank k for bottom-rank approximation impact the practical deployability of the edit reversal method?
- How can the reversal method be made less model-dependent?
- To what extent does the non-identical nature of the model's reversed outputs (despite being semantically similar) and the accuracy imbalance between edit removal and original output recovery limit the utility of the reversal method?

---

> ### Author Response · Authors · 2025-11-20
>
> We appreciate the reviewer’s comments and are pleased that they found our work to offer practical, efficient, and high-performing methods.
>
> > W1: The effectiveness of the reversal (and analysis of rank-one approximations) is shown to be model-dependent. For example, the optimal rank k for bottom-rank approximation varies significantly across models (e.g., k=11 for GPT2-XL vs. k=15 for llama3 to achieve highest reversal accuracy), and the similarity of the top rank-one approximation to the update matrix is much lower for LLAMA3 than for GPT models. This suggests a need for model-specific tuning of the reversal hyperparameter.
>
> > Q1: How does the required model-specific tuning of the optimal rank k for bottom-rank approximation impact the practical deployability of the edit reversal method?
>
> > Q2: How can the reversal method be made less model-dependent?
>
> The reversal approach does perform differently across models. However, these differences in performance do not undermine the practicality of our approach, since bottom-rank approximations with $k=15$ as a default value provide strong reversal performance across all evaluated models. The absolute differences in performance between the optimal $k$ value and $k=15$ are rather minor (0.64 p.p. for GPT2-XL, 1.29 p.p. for GPT-J, 0 p.p. for LLAMA since $k=15$ is the optimal value). Having this default value for reversal would make the approach less model-dependent. For optimal performance, one could tune a model-specific $k$ value on a validation set.
>
> > W2: The core of the methods, especially the reversal technique, relies on the assumption that the malicious edit is a rank-one update (like ROME or r-ROME). The effectiveness for different types of model edits is not explored. If the proposed method only works on rank-one update methods, the usability is limited since there are many other types of model editing methods like memory-based and constrained-tuning-based methods.
>
> We added results on other editing methods: MEND, MEMIT, and AlphaEdit (see Appendix A in the revised version). Our results show strong generalization. One exception is tracing edits with AlphaEdit, where we observe low performance due to AlphaEdit avoiding overfitting to the edited object by projecting its changes onto the null space.
>
> > W3: While the reversal accuracy is high (up to 93%). A qualitative analysis showed that even when reversed, the outputs, while semantically similar, are sometimes not identical to the original model's unedited outputs. Also, the decrease in editing accuracy is higher than the increase in reversal accuracy, indicating the method is better at removing the edit than fully recovering the original output distribution.
>
> > Q3: To what extent does the non-identical nature of the model's reversed outputs (despite being semantically similar) and the accuracy imbalance between edit removal and original output recovery limit the utility of the reversal method?
>
> The aim of our reversal approach is to neutralize malicious edits. In this context, producing a non-malicious output that does not contain the edit and that semantically aligns with the original outputs is rather a minor issue compared to a malicious output, and the fact that they are not identical poses minimal practical limitations.
>
>
> To further validate this, we  assess whether these minor output differences have any downstream impact. We evaluate 310 reversed models on six tasks from the GLUE benchmark. The results (see Figure 5 and L402-L426 in the revised version) show that the reversed models perform on par with the edited models before reversal (average F1-score of 55.18% for the edited models vs. 55.49% for the reversed models), indicating that reversal does not have any negative effects on the model’s performance.

---

> > ### Author Response · Authors · 2025-11-28
> >
> > Thank you for your thoughtful feedback on our work. We have updated the paper to include results on reversing sequential edits in Appendix C (Figures 10 and 11). Similar to the batch editing setting, reversal with AlphaEdit maintains single-edit performance levels, while reversal with MEMIT suffers a performance drop. We would be happy to answer any questions you might have.

---

### Official Review · Reviewer_AH7s · 2025-11-01

**Soundness:** 3
**Presentation:** 2
**Contribution:** 2
**Rating:** 6
**Confidence:** 3

**Summary:**

The paper proposes methods to detect and undo malicious or unintended knowledge edits. These edits modify a model’s internal parameters to change factual outputs, which can be useful for updating information but also pose security risks. The authors introduce two tasks: tracing edits and reversing edits. Their tracing method infers the edited object solely from altered weights, achieving high accuracy. Their reversal method uses bottom-rank approximations from singular value decomposition to remove edits without retraining. The study demonstrates strong generalization across 2 datasets and 4 models, showing that both tracing and reversal are feasible using only model weights.

**Strengths:**

- The paper introduces a training-free framework for detecting and reversing malicious edits directly from model parameters, a new defense direction for LLM safety.
- Experimental results show high accuracy and generalization across different models and datasets, suggesting good robustness.
- The methods are computationally efficient and require no access to original weights or editing prompts, enhancing practical applicability for security auditing.

**Weaknesses:**

- The study focuses only on rank-one edits, limiting applicability to other editing methods and scenarios like MEMIT, MEND, SERAC.
- The motivation
The evaluation scope is restricted to controlled datasets and synthetic edits, leaving real-world validation uncertain.
- The interpretability of why bottom-rank approximations work well for reversal is not fully explored, reducing theoretical clarity of the mechanism.
- The motivation for reversing edits is questionable, since model editing is primarily designed to update or add new knowledge rather than to be undone; thus, the practical need for edit reversal appears limited. The paper does not compare its reversal method against simply reapplying the inverse or original edit, which would be a more direct and intuitive baseline for restoring model behavior

**Questions:**

- Can you clarify whether the proposed tracing and reversal techniques would still work, or how they might adapt, when applied to higher-rank or alternative editing methods such as MEMIT, MEND, or SERAC?
- Have you considered testing their approach on real or naturally occurring edits (beyond controlled datasets) to assess its reliability in more practical or adversarial settings?
- Can you provide a clearer explanation or empirical evidence for why bottom-rank components capture “pre-edit” information and effectively reverse edits?
- What is the practical advantage of using the proposed reversal method over simply reapplying the inverse or original edit, especially given that model editing is typically meant to add or update knowledge rather than undo it?

---

> ### Author Response · Authors · 2025-11-20
>
> We thank the reviewer for their feedback and are encouraged that they view our work as introducing a new direction for LLM safety through robust and practical methods.
>
> > W1: The study focuses only on rank-one edits, limiting applicability to other editing methods and scenarios like MEMIT, MEND, SERAC.
> > Q1: Can you clarify whether the proposed tracing and reversal techniques would still work, or how they might adapt, when applied to higher-rank or alternative editing methods such as MEMIT, MEND, or SERAC?
>
> We added results on other editing methods such as MEND, MEMIT and AlphaEdit (see Appendix A and Table 5, 6, 9 and Figure 9 in the revised version). Our results show strong generalization. One exception is tracing edits with AlphaEdit, where we observe low performance due to AlphaEdit avoiding overfitting to the edited object by projecting its changes onto the null space.
>
> > W2: The motivation The evaluation scope is restricted to controlled datasets and synthetic edits, leaving real-world validation uncertain.
> > Q2: Have you considered testing their approach on real or naturally occurring edits (beyond controlled datasets) to assess its reliability in more practical or adversarial settings?
>
> Since our work builds on editing methods, we benchmarked datasets on which these editing methods were evaluated and were shown to have good performance. In order to consider other datasets, first we need to confirm that the editing methods still perform well on these datasets, which we believe is beyond the scope of our work.
>
> > W3: The interpretability of why bottom-rank approximations work well for reversal is not fully explored, reducing theoretical clarity of the mechanism.
>
> > Q3: Can you provide a clearer explanation or empirical evidence for why bottom-rank components capture “pre-edit” information and effectively reverse edits?
>
> Our analysis in Section 6.2 and Figure 4 shows that the update matrices have a relatively high similarity  to the top few rank-one approximations of the edited matrices, i.e., the edit is encoded in the top few rank-one approximations. This motivates our reversal approach, which is based on excluding the top few rank-one approximations using bottom-rank approximations. Our results in terms of reversal accuracy, editing accuracy (Table 2) and KL-divergence (Table 4) confirm that the model is able to retrieve the original outputs (or semantically similar outputs) after replacing the edited matrices with bottom-rank approximations. It is also worth noting that when using bottom-rank approximations we are keeping the majority of the approximations (e.g., in case of LLAMA3 we are keeping 4081 out of the original 4096 approximations, i.e., 99.6%), which should encode the majority of the model’s original knowledge.
>
> > W4: The motivation for reversing edits is questionable, since model editing is primarily designed to update or add new knowledge rather than to be undone; thus, the practical need for edit reversal appears limited. The paper does not compare its reversal method against simply reapplying the inverse or original edit, which would be a more direct and intuitive baseline for restoring model behavior
>
> > Q4: What is the practical advantage of using the proposed reversal method over simply reapplying the inverse or original edit, especially given that model editing is typically meant to add or update knowledge rather than undo it?
>
> Indeed, model editing is designed to update knowledge in LLMs. However, recent work shows that model editing can be used maliciously, and mentions reversing edits as a potential remedy [1], which is the main motivation for our work.
>
> The main advantage of our approach compared to re-applying the inverse edit is that our method does not assume knowing the edited knowledge, which would be required for re-applying the inverse edit. Instead, our method operates directly on the edited weights without any information about the edit. Additionally, our method is training-free and therefore more efficient than editing the model.
>
> [1] Position: Editing Large Language Models Poses Serious Safety Risks (Youssef et al., ICML 2025)

---

> > ### Author Response · Authors · 2025-11-28
> >
> > Thank you for your thoughtful feedback on our work. We have updated the paper to include results on reversing sequential edits in Appendix C (Figures 10 and 11). Similar to the batch editing setting, reversal with AlphaEdit maintains single-edit performance levels, while reversal with MEMIT suffers a performance drop. We would be happy to answer any questions you might have.

---

### Official Review · Reviewer_8BSp · 2025-11-01

**Soundness:** 2
**Presentation:** 3
**Contribution:** 3
**Rating:** 6
**Confidence:** 3

**Summary:**

This paper studies how to trace and reverse malicious edits in large language models that were modified using rank-one editing methods like ROME or r-ROME. It introduces two complementary defenses: (1) Tracing, which identifies the specific edited fact (object) directly from the edited weight matrix without needing the original weights or prompts, and (2) Reversing, which removes the malicious change by replacing the edited matrix with its bottom-rank singular value decomposition (SVD) approximation, effectively removing the edit signal concentrated in the top singular modes. Experiments across multiple LLMs show that tracing achieves near-perfect accuracy and reversal restores the model’s original behavior with high fidelity.

**Strengths:**

+ The proposed defense is practical and lightweight, requiring only access to the edited weights and no training data or edit prompts, making it suitable for real-world forensic use.
+ The reversal approach is simple yet effective, using an interpretable SVD-based method that efficiently removes the edit signal while maintaining model integrity.
+ The experimental validation is comprehensive and convincing, demonstrating strong performance across multiple models and datasets with clear quantitative results and ablation studies.

**Weaknesses:**

- The method’s generality is limited, as it is evaluated only on single-layer, rank-one edits and may not extend to more complex, multi-layer, or non-rank-one scenarios.
- The evaluation scope is narrow, focusing mainly on object recovery and KL divergence without exploring broader behavioral or capability effects after reversal.

**Questions:**

See the weaknesses.

---

> ### Author Response · Authors · 2025-11-20
>
> We thank the reviewer for their comments and suggestions and are encouraged that they found our work to offer practical and efficient defense methods supported by comprehensive and convincing evaluation.
>
>
> > The method’s generality is limited, as it is evaluated only on single-layer, rank-one edits and may not extend to more complex, multi-layer, or non-rank-one scenarios.
>
>
> We added results on other editing methods such as MEND, MEMIT and AlphaEdit (see Appendix A and Table 5, 6, 9 and Figure 9 in the revised version). Our results show strong generalization. One exception is tracing edits with AlphaEdit, where we observe low performance due to AlphaEdit avoiding overfitting to the edited object by projecting its changes onto the null space.
>
>
> > The evaluation scope is narrow, focusing mainly on object recovery and KL divergence without exploring broader behavioral or capability effects after reversal.
>
>
> In order to investigate any potential effects of our reversal approach on model capabilities, we added an evaluation on downstream tasks. More specifically, we evaluated LLAMA3 before and after reversal on six tasks from the GLUE benchmark. The results (see Figure 5 and L402-L426 in the revised version) show that the reversed models perform on par with the edited models before reversal (average F1-score of 55.18% for the edited models vs. 55.49% for the reversed models), indicating that reversal does not have any negative effects on the model’s performance.

---

> > ### Author Response · Authors · 2025-11-28
> >
> > Thank you for your thoughtful feedback on our work. We have updated the paper to include results on reversing sequential edits in Appendix C (Figures 10 and 11). Similar to the batch editing setting, reversal with AlphaEdit maintains single-edit performance levels, while reversal with MEMIT suffers a performance drop. We would be happy to answer any questions you might have.

---

### Author Response · Authors · 2025-12-03

We thank all reviewers for their valuable feedback. We believe we addressed all of their concerns during the author-response period with extended experiments in the revision. We provide a summary of the reviewers’ comments and our responses here (please refer to our direct responses to the reviews for more details):


| Weakness/Question  | Author Response  |
| :---- | :---- |
| \[Extending Experiments\] Evaluation is only on rank-one methods.  | We added experiments for MEMIT, AlphaEdit and MEND (Appendix A). Results show that our approach for *tracing* and *reversing* generalizes beyond rank-one methods, except for *tracing* edits with AlphaEdit, where we observe low performance due to AlphaEdit avoiding overfitting to the edited object by projecting its changes onto the null space.  |
| `W1` from `8BSp` |  |
| `W1` and `Q1` from `AH7s` |  |
| `W2` from `aKZH` |  |
| `W2` from `QZJb` |  |
|  |  |
| \[Extending Experiments\] Extending evaluation to batch edits and sequential edits.  | We added experiments for reversing edits in a batch editing setting (Appendix B, Figure 8 and 9), and a sequential editing setting (Appendix C, Figure 10 and 11). In both settings, reversal with AlphaEdit maintains single-edit performance levels, while reversal with MEMIT suffers a performance drop.  |
| `W3` from `aKZH` |  |
|  |  |
| \[Extending Experiments\] Evaluation misses effects on model capabilities after reversal.  | We added an experiment to evaluate the general capabilities of the reversed models on 6 tasks from the GLUE benchmark. Results showed that the reversal process does not negatively affect the models (Section 6.3 L402-426). |
| `W2` from `8BSp` |  |
| `W3` and `Q3` from `QZJb`  |  |
|  |  |
| \[Extending Experiments\] No real world evaluation. | Since our work is based on editing methods, we used the benchmark datasets on which these editing methods were evaluated and were shown to have good performance in prior work. In order to consider other datasets, we first need to confirm that the editing methods still perform well on these datasets, which we believe is beyond the scope of our work. |
| `W2` and `Q2` from `AH7s` |  |
|  |  |
| \[Clarification\] Why/How do bottom-rank approximations lead to reversal (empirical evidence)?  | Our analysis in Section 6.2 and Figure 4 shows that the update matrices have a relatively high similarity to the top few rank-one approximations of the edited matrices, i.e., the edit is encoded in the top few rank-one approximations. This motivates our reversal approach, which is based on excluding the top few rank-one approximations using bottom-rank approximations. Our results in terms of reversal accuracy, editing accuracy (Table 2\) and KL-divergence (Table 4\) confirm that the model is able to retrieve the original (or semantically similar) outputs after replacing the edited matrices with bottom-rank approximations, i.e., excluding the editing signal leads to recovering the original (or semantically similar) outputs.  |
| `W3` and `Q3` from `AH7s` |  |
|  |  |
| \[Clarification\] Motivation for reversal, why do you not re-apply the inverse edit?  | Recent work shows that model editing can be used maliciously, and suggest reversing edits as a potential remedy \[1\]. This is the main motivation for our work. Re-applying the inverse edit would assume *knowing* the original edit, which is an unrealistic assumption for malicious use cases. Our approach operates directly on the edited weights without any information about the edit. Additionally, our method is training-free and therefore more efficient than re-editing the model. |
| `W4` and `Q4` from `AH7s`  |  |
|  |  |
| \[Clarification\] The need for model-specific tuning of the $k$-value for reversal. How to make the reversal method less model-dependent?  | The differences in performance across models do not undermine the practicality of our approach, since bottom-rank approximations with $k=15$ as a default value provide strong reversal performance across all evaluated models. With this strong default, reversal is not model-dependent. For optimal performance, one could tune a model-specific value on a validation set.  |
| `W1`, `Q1` and `Q2` from `QZJb`  |  |
|  |  |
| \[Clarification\] Clarifying the perceived discrepancy between the analysis in Figure 4, and the results in Table 2  | We use **rank-one** approximations in the analysis in Figure 4 to motivate the approach, and use **bottom-rank** approximations for reversal in Table 2\. We added results for Qwen in Figure 4\. |
| `W4`, `Q2-6` from `aKZH` |  |

---

### Meta-Review · Area_Chair_SioE · 2026-01-05

**Summary:**

This paper proposed a method to infer the edited object entity based on the modified weights, and also an effective and training-free method for reversing edits.

Reviewers initially raised the following concerns:
(1) method generality beyond rank-one edits,
(2) evaluation scope, particularly behavioral or capability effects after reversal,
(3) lack of evaluation on real-world edits,
(4) insufficient analysis of why bottom-rank approximations enable reversal,
(5) motivation for reversing edits,
(6) practicality and robustness of the method,
(7) incomplete recovery of the original output distribution after reversal,
(8) writing clarity, and
(9) seemingly contradictory empirical results.

After the rebuttal and revisions, the AC believes that all major concerns have been adequately addressed.

For (1), the authors added experiments on additional editing methods, including MEND, MEMIT, and AlphaEdit. The method generalizes well to MEND and MEMIT. The lower performance on AlphaEdit is clearly explained by its design choice, and the explanation is reasonable and acceptable.

For (2), the authors evaluated downstream model performance after reversal, demonstrating that the reversal process does not negatively affect model capabilities.

For (3), although real-world edits are not evaluated, the authors’ justification for using established benchmark datasets is reasonable, and the current evaluation remains appropriate for the scope of this work.

For (4), the authors provided additional analysis, which is reasonable to AC.

For (5), the authors explained it in the rebuttal, which is reasonable to AC

For (6), the authors demonstrated that the method shows a good performance with a default setting (e.g., k = 15) and the absolute differences in performance between the optimal value and k=15 are minor.

For (7), the authors conducted further experiments showing that reversed models perform on par with edited models prior to reversal.

For (8) and (9), the authors provided detailed clarifications and revisions that resolved concerns about writing clarity and the perceived contradictions in the empirical results.

Overall, this work presents a solid and practically relevant contribution to LLM safety and interpretability, with thoughtful responses to reviewer feedback and strengthened empirical support. AC tends to accept this paper.

**Reviewer Concerns:**

Reviewers initially raised the following concerns:
(1) method generality beyond rank-one edits,
(2) evaluation scope, particularly behavioral or capability effects after reversal,
(3) lack of evaluation on real-world edits,
(4) insufficient analysis of why bottom-rank approximations enable reversal,
(5) motivation for reversing edits,
(6) practicality and robustness of the method,
(7) incomplete recovery of the original output distribution after reversal,
(8) writing clarity, and
(9) seemingly contradictory empirical results.

After the rebuttal and revisions, the AC believes that all major concerns have been adequately addressed.

For (1), the authors added experiments on additional editing methods, including MEND, MEMIT, and AlphaEdit. The method generalizes well to MEND and MEMIT. The lower performance on AlphaEdit is clearly explained by its design choice, and the explanation is reasonable and acceptable.

For (2), the authors evaluated downstream model performance after reversal, demonstrating that the reversal process does not negatively affect model capabilities.

For (3), although real-world edits are not evaluated, the authors’ justification for using established benchmark datasets is reasonable, and the current evaluation remains appropriate for the scope of this work.

For (4), the authors provided additional analysis, which is reasonable to AC.

For (5), the authors explained it in the rebuttal, which is reasonable to AC

For (6), the authors demonstrated that the method shows a good performance with a default setting (e.g., k = 15) and the absolute differences in performance between the optimal value and k=15 are minor.

For (7), the authors conducted further experiments showing that reversed models perform on par with edited models prior to reversal.

For (8) and (9), the authors provided detailed clarifications and revisions that resolved concerns about writing clarity and the perceived contradictions in the empirical results.

**Reviewer Scores:**

AC thinks reviewer QZjb and aKZHwill increase their score after reading the rebuttal. Authors addressed these concerns well.

---

### Decision · Program_Chairs · 2026-01-26

Accept (Poster)